# Laser Transmission Welding of Semi-Crystalline Polymers and Their Composites: A Critical Review

**DOI:** 10.3390/polym13050675

**Published:** 2021-02-24

**Authors:** Foram Dave, Muhammad Mahmood Ali, Richard Sherlock, Asokan Kandasami, David Tormey

**Affiliations:** 1Department of Mechanical and Manufacturing Engineering, Institute of Technology Sligo, Ash Lane, F91 YW50 Sligo, Ireland; foram.dave@mail.itsligo.ie (F.D.); Ali.MuhammadMahmood@itsligo.ie (M.M.A.); 2Centre for Precision Engineering, Materials and Manufacturing (PEM) Centre, Institute of Technology Sligo, Ash Lane, F91 YW50 Sligo, Ireland; sherlock.richard@itsligo.ie; 3Department of Life Sciences, School of Science, Institute of Technology Sligo, Ash Lane, F91 YW50 Sligo, Ireland; 4Materials Science Group, Inter-University Accelerator Centre, Aruna Asaf Ali Marg, New Delhi 110 067, India; asokan@iuac.res.in or

**Keywords:** Industry 4.0, laser transmission welding, semi-crystalline polymers, composites, carbon black, characterisation, laser welding of polymers, laser power, scanning speed, clamping pressure

## Abstract

The present review provides an overview of the current status and future perspectives of one of the smart manufacturing techniques of Industry 4.0, laser transmission welding (LTW) of semi-crystalline (SC) polymers and their composites. It is one of the most versatile techniques used to join polymeric components with varying thickness and configuration using a laser source. This article focuses on various parameters and phenomena such as inter-diffusion and microstructural changes that occur due to the laser interaction with SC polymers (specifically polypropylene). The effect of carbon black (size, shape, structure, thermal conductivity, dispersion, distribution, etc.) in the laser absorptive part and nucleating agent in the laser transmissive part and its processing conditions impacting the weld strength is discussed in detail. Among the laser parameters, laser power, scanning speed and clamping pressure are considered to be the most critical. This review also highlights innovative ideas such as incorporating metal as an absorber in the laser absorptive part, hybrid carbon black, dual clamping device, and an increasing number of scans and patterns. Finally, there is presented an overview of the essential characterisation techniques that help to determine the weld quality. This review demonstrates that LTW has excellent potential in polymer joining applications and the challenges including the cost-effectiveness, innovative ideas to provide state-of-the-art design and fabrication of complex products in a wide range of applications. This work will be of keen interest to other researchers and practitioners who are involved in the welding of polymers.

## 1. Introduction

Thermoplastics have been used widely in manufacturing industries, especially in the medical and other industries. Joining thermoplastics, in small scale products such as medical devices or electronics, as well as in large scale products such as the automobile, aerospace, or construction industries has been considered as challenging and onerous to achieve the best quality products [1]. Therefore, joining technology has been growing and developing the field to achieve different requirements, such as temperature resistance, mechanical strength, or media resistance [2]. The primary interest lies in the suitable methods and options for joining two surfaces for specific or general applications, simulating and optimising the process [3,4,5,6]. Fortunately, a laser (an acronym of light amplification by stimulated emission of radiation) is one of the best ways, to date, to fulfil these requirements. It has never been envisaged that a single tool laser would revolutionise the world, due to its extensive use in industries, medical diagnostics and telecommunication [7]. In material processing and manufacturing, especially in joining (weld) technology, lasers have been widely adopted by industries along with their technological developments. In the early 1970s, a CO_2_ laser was introduced for direct laser welding [8]. Thereby, the lasers were demonstrated as being suitable for welding, especially plastics [9,10]. However, this process was considered expensive in competition with the conventional technologies of welding the components. In 1985, laser welding using Nd: YAG laser was carried out and patented by Toyota (Japan) to modify the absorption properties of the laser absorptive (LA) polymer using carbon black (CB) additive [11] and later diode and fibre lasers were introduced. Since the middle of the 1990s, laser technology has been widely accepted as a manufacturing tool due to its high speed, non-contact and precision with low heat effects and easy access of various laser types such as gas (CO_2_) lasers, solid-state (Nd-YAG) lasers, semiconductor (diode) lasers, and fibre lasers [12]. In electronic packaging, automotive and medical applications, plastic welding using continuous wave (CW) plays an important role. Developments have been made in joining textile and other dissimilar materials such as plastics to ceramics or metals [13,14].

Lasers have also been playing a key role in achieving the requirements of the fourth industrial revolution, often referred to as Industry 4.0, which branches the technologies for underpinning information transparency, technical assistance, inter-connectivity, and decentralised decisions [15,16]. Frank et al. [15] proposed a conceptual framework of Industry 4.0 technologies, as shown in Figure 1. The framework has been classified into two main layers: the front-end and the base technologies. Smart manufacturing, smart products, smart supply chain and smart working are categorised under the first layer (front-end technologies), each of them represents a specific subset of techniques. Technologies such as cloud computing, internet of things (IoT), which provide connectivity and intelligence to the first layer or the front-end technologies, are termed as the base technologies or the second layer. Smart manufacturing is the central element of this concept with enhanced automation, and the smart product is its extension. The transformation of the manufacturing activities based on emerging technologies (smart manufacturing) and the way products are offered (smart products) are considered in front-end technologies [17]. Chiarello et al. [18] divided the technologies in Industry 4.0 into various clusters. Lasers are considered under the production cluster (cluster-10), as highlighted in Figure 2. There are precise placements of welds facilitating automation, and systematisation in the production lines [19,20,21,22,23]. Precise welding of hard-to-reach areas can be possible using this technique under robotic control [24,25,26].

Various types of laser welding are available based on; laser source: continuous, pulsed, solid-state, gas, diode, and fibre; the geometry of materials: butt, corner, edge, lap, and T-joint; nature of the interaction between laser and material: direct, surface heating, through transmission; and laser beam delivery: contour, simultaneous, quasi-simultaneous and masked [27,28,29,30]. Generally laser sources such as Nd: YAG, diode and fibre lasers, can be selected based on operating wavelength suitable for LTW of polymers and their composites. The range of wavelengths include for Nd: YAG as 1064 nm, for diode lasers as 405–1940 nm, for fiber lasers as 1000–2100 nm and CO_2_ laser as 10.6 µm [31]. This review will focus on Laser Transmission Welding (LTW) of polymers based on laser diode CW (commercial welding application) and lap joint welding using CB as filler. Diode laser welding results in cleaner and stronger joints [32]. The diode LTW consists of two polymer parts where the beam passes through the laser transmissive (LT) polymer and dissipates heat in the LA layer, resulting in the joining of two surfaces as a result of melting and inter-diffusion of polymer chains [33]. Several factors are influencing the quality of the weld, as shown in Figure 3. Therefore, it is essential to have the optimised laser welding process to save time and other resources used in the manufacturing process.

### 1.1. Laser Transmission Welding (LTW)

LTW of polymers is a joining technique frequently selected by contemporary researchers to weld two thermoplastic surfaces, as illustrated in Figure 4. LTW avoids the generation of byproduct or particle release, unlike adhesive bonding [34]. Diode laser for LTW is considered due to its capability to encapsulate sensitive electronic housings without any damage compared to other techniques such as ultrasonic welding. Furthermore, LTW is a non-contact and non-contaminant process involving an optically transparent part (LT) for a given laser λ to pass through and an absorbing surface (LA) that melts and forms a hermetic sealing [35,36,37,38,39]. Preferential deposition of energy is required in the interfacial zone, which eventually becomes the weld zone. The surrounding area of the weld zone, which is affected by residual heat, is considered as the heat-affected zone (HAZ). In LTW, there is no vibration of the components, and therefore, the welding of materials from thin sheets (~0.01 mm) to thick plates (~50 mm) with high performance, good flexibility, and low distortion can easily be achieved [40].

Overlap joint assembly is the most basic configuration for LTW of polymers, as shown in Figure 4 [41,42,43]. The laser energy transmitted through LT is converted into heat with the help of LA, which melts the material. The clamping mechanism is used to make the layers in intimate contact and in this way, heat is conducted through LA to LT [24,44]. After inter-diffusion of the material, the joint is created, and this welding technique is used in almost all thermoplastics. The laser-welded products, specifically continuous reinforced thermoplastic composites [40], provide a good stiffness and strength level besides other properties like chemical resistance [26]. However, to achieve a strong joint or weld integrity, it is highly essential to understand the laser process parameters’ specifications and capability, factors leading to weld defects, welding mechanism, and the materials’ properties [45].

### 1.2. LTW of Polymers and Their Composites

Elastomers and thermosets exhibit soft elasticity and hard elasticity, respectively, and cannot be melted. Therefore, thermoplastics are preferred in industrial applications where laser welding is required. Furthermore, thermoplastics can be divided into two types, such as amorphous and semi-crystalline thermoplastics. The present study is focused on semi-crystalline (SC) polymers, and the most common is semi-crystalline polypropylene (PP). PP is nontoxic, lightweight and easy in forming shapes with varying geometry, and it has gained enormous interest in LTW applications [46].

Several experimental techniques have also been discussed, which characterises the polymer-filler interactions and the degree of filler dispersion in the polymer matrix [47]. Various authors have studied the influence of laser welding parameters on polymer morphology [48]. Ghorbel et al. [49] investigated the effect of process parameters using 120 W diode laser at 940 nm, both the laser power (20–40 W) and the scanning speed (3–6 mm/s) on the geometry and the microstructure of the diode laser-welded PP. The increase of the laser power and the decrease of the scanning speed are found to result in a larger volume of the weld zone and depth of penetration. The laser’s thermal energy gets trapped within the polymers and may lead to a rise in temperature in deeper regions [50]. Higher values of laser power and low scanning speed can degrade the morphological and optical properties of polymers [51]. Prabhakaran et al. [52] carried out analysis using CW diode laser of maximum power 150 W and a focal length of 100 mm. They reported that low laser power with lower speed results in maximum weld strength. Optimising a good weld quality parameter is a challenging task that involves various trial and errors and a detailed investigation and analysis.

The ultimate goal for any industry is to increase productivity with minimum scrap rate. Different types of lasers are used with increased laser power and scanning speeds [26,41]. Applying simulation, mathematical modelling [53,54,55], optimisation [56], monitoring and adaptive control of LTW processes [57] for industrial products are some of the recent advancements made in this field.

Besides, in most industrial applications, polymers are used as composite materials due to their better thermo-mechanical properties. For instance, CB filler incorporated into polymer matrix imparts useful properties such as reinforcement, tensile strength, tensile modulus, flexural strength and flexural modulus [58]. These properties depend upon the type of CB (shape, size, surface area, surface activity, aggregate structure, etc.), dispersion and distribution of the filler and filler-polymer interactions [59]. There is a need to focus on compounding ingredients, especially CB and salient processing parameters on the polymer morphology, which ultimately controls the laser energy and amount of melting along with the prominent laser welding parameters. Understanding the melting and inter-diffusion phenomena during the formation of the weld joint is significantly essential. The literature on the weld characterisation techniques and the recent advances in laser processing is available in a discrete manner. Hence, this article highlights the current status of the LTW of SC polymers and their composites. It also includes discussing the importance of CB dispersion to the laser-welded product performance and its economic aspects. The diode laser transmission welded SC polymers are to be characterised and analysed to understand the pre and post-processing techniques (Table 1).

## 2. Laser Interaction with Polymer

LTW has been known for high precision manufacturing and polymer processing of complicated and micro geometries [13]. The laser beam of high energy is made to focus on a confined region of the polymer for achieving a desirable response. For polymers that are which are opaque to these radiations, laser energy only results in the surface modifications such as altering the surface chemistry or altering the crystal structure of the initial few layers of polymer surface when laser interacts for a short period. This could be in the case of highly crystalline polymers. The initial few layers of polymer melts and the crystal structure gets altered. Laser interaction leading to surface modification is unique, causing permanent changes in the polymer’s properties such as resistance to wear and surface damage. Surface chemistry and morphology alteration find application in biomaterials for enhancing adhesion of proteins and cells for the lab-on-a-chip type of sensors and biomedical implants [70].

Figure 4 shows the hypothetical steps involved in the interaction of the laser beam during the LTW process of a SC polymer consisting of amorphous and crystalline regions. During LTW, there can be tiny bubbles in the polymer matrix during the welding process. The expansion pressure of the bubbles formed would push the polymer in the molten form to diffuse into one another [13]. In the current article, we have tried to provide a stepwise explanation of the changes taking place (morphological, thermal, optical and mechanical) as the laser interacts with the polymer (specifically SC PP). However, it would be interesting to understand the chemical changes occurring in polymers such as PP, which consists of carbon and hydrogen atoms.

In the case of LTW, as the laser beam is focused initially on the polymer surface, the energy of the laser beam (photons) interacts with the virgin polymer at room temperature. The temperature of the LT increases, and the laser beam continues its interaction with the remaining portion of the polymer bulk. A certain portion of the laser beam gets reflected (*R*) from the polymer interface depending upon the polarisation (s-polarised or p-polarised components of the radiation), angle of incidence and the refractive index of the atmosphere *n*_1_ and the material *n*_2_. For the incident light to be normal on the flat polymer surface, *R* is given by [71]:(1)$$R=R_{s}=R_{p}={\left(\frac{n_{1}-n_{2}}{n_{1}+n_{2}}\right)}^{2}$$

As the laser beam enters the polymer matrix, the intensity gets decayed along with the depth as it transmits and gets absorbed by the other joining part. The decay rate depends upon the absorption coefficient of the material *α* (1/m), which is a function of *λ* (m) and temperature (K). According to the Beer–Lambert law, for a constant value of α, the decay of the laser intensity *I* (W/m^2^) is exponential to the depth z (m) [71]
(2)$$I\;\left(z\right)=I_{o}\;e^{-\alpha z}$$
where *Io* is the intensity of the radiations just inside the polymer surface after reflection loss. The intensity of the transmitted radiations decreases with depth of δ = 1/*α* (m). At absorption depth δ, the laser intensity drops to 1/e of its initial value. The depth of penetration or the level of bulk modification can be controlled by choosing the *λ* with desired absorption depths.

It is important to understand the principle and equations defining the laser beam absorption and heat transfer in the polymer bulk [70]. The heat equation can be derived from energy conservation and the Fourier’s law of heat conduction. Accordingly, the heat transfer rate through a given material is directly proportional to the negative gradient of the temperature. In a defined coordinate system for a laser beam, the heat equation is given by [72]:(3)$$\rho \left(x,\;T\right)c_{p}\left(x,\;T\right)\frac{\partial T\left(x,\;t\right)}{\partial t}\;-\;\nabla \;\left[k\left(x,\;T\right)\nabla T\;\left(x,\;t\right)\right]+\;\rho \left(x,\;T\right)c_{p}\left(x,\;T\right)v_{s}\nabla T\;\left(x,t\right)=\;Q\;\left(x,t\right)$$
where:

ρ: mass density (kg/m^3^)*c_p_*: specific heat at constant pressure (J /(kg·K))*k*: thermal conductivity (W/(m·K))*v_s_*: velocity of the substrate relative to the source of heat (m/s)

Above equation consists of conductive as well as convective (vs) terms of heat transfer. The value of *k* can be increased by the type and dosage of CB in the polymer matrix. The polymer morphology formed as a result of polymer processing conditions, type of compounding ingredients, its dispersion and distribution in the polymer matrix determines the thermal diffusion length of the laser beam (*l_T_*) in metre (m). It is defined as [70]:(4)$$l_{T}\approx \;\zeta \;\sqrt{D\tau }$$
where:D (=λ/𝜌 *c_p_*): thermal diffusivity of the polymer (m^2^/s)ζ: geometric constantτ: the characteristic time (s)

The polymer matrix temperature changes as it propagates into the bulk in a given characteristic time τ. As the radiation propagates beyond *l_T_* into the LA, the temperature of the polymer matrix rises. The morphological, optical, thermal and mechanical properties of the polymer changes as the heat energy spread during and after the laser beam radiate and propagate into the polymer. Figure 4 shows the sequence of laser interactions with the LT and LA SC polymer. The highlighted number (6) in Figure 4 shows the polymer interface where the various phenomena occur (surface rearrangement and approach, surface wetting, diffusion and randomisation), leading to weld zone formation. HAZ (8, 9 and 10 in Figure 4) may have significant differences as compared to the other part of the polymer matrix. Figure 4 shows the three zones (8, 9 and 10) of HAZ formed around the weld joint. The morphological differences between these three zones will be explained in detail in Section 3.2.

The joining phenomenon of two polymers involves the interfacial reaction between the two surfaces. Due to the low surface tensions in polymers, they have poor wettability. This could be improved through polymer surface excitation, corona discharge, thermal simulation of molecular mobility, etc. The melting and inter-diffusion phenomenon taking place during the LTW process and the role of various compounding ingredients (specifically CB), polymer processing conditions, laser parameters, etc., will be discussed further in detail.

## 3. Effect of Compounding Ingredients on the Properties of Semi-Crystalline Polymer

The nucleating agents or pigments in the polymer matrix influence the welding process and weld quality. Suitable nucleating agents are to be chosen to ensure the transmission and absorption of the laser beam. These nucleating agents modify the material’s thermal, mechanical, and optical properties [73] depending upon their type and concentration and ultimately decide the process window for welding two thermoplastics. No welding would occur for such unpigmented polymers in the case of material containing no fillers or pigments in the polymer matrix [10]. There should be an absorbance at the interface which allows welding to be carried out. Using various additives such as CB [74], the absorption coefficient of polymers can be increased. The CB will absorb the laser beam and resonate directly at the photon energy of the laser. There is a small percentage of scattering of the laser beam leading to an increased bulk absorption.

### 3.1. Effect of Fillers in the Laser Absorptive Layer

#### 3.1.1. Carbon Black (CB)

CB is one of the compounding ingredients used in a polymer formulation mainly to enhance the reinforcement of the final product. CB is spherical shaped elemental carbon that is coalesced into colloidal size aggregates. It is an intermediate form of matter with a definite two-dimensional repetition within each layer, even though these are not amorphous [75], but a particular class of mesomorphic solids having a unique name as “turbostratic (unordered layers). Each turbostratic group consists of roughly stacked graphite layers but with completely random orientation within each layer.

These CB are synthesised by thermal degradation or incomplete combustion of hydrocarbons [76]. During the production of CB, the primary particles that are formed rarely exist in an isolated form. The particles are fused, leading to the formation of aggregates of various sizes and shapes. The aggregates are held together by weak van der Waals forces or by binders [77]. These aggregates are known as “structure” [78]. These structures get entangled and bound together physically, known as agglomerates. CB’s properties, such as density, particle size, BET (Brunauer, Emmett and Teller), oil absorption number (OAN), tint strength, etc. are essential to know for selecting the type of CB for a particular formulation. These properties are determined by aggregate size distribution and morphology. The principle features are the size and shape of CB in the composite systems, which determine CB’s performance as a reinforcing agent and as a pigment [77]. Smaller particle size or larger surface area results in higher tensile strength [79]. These properties, as well as the processing conditions of polymer composites, ultimately determine the weld strength of the laser-welded composites.

The shape of the filler plays a significant role in the performance of polymer composites [80]. It was observed that composites containing larger and spherical shape particles were much easier to be mixed due to the small filler surface area. The larger surface area of the smaller filler particles requires more polymer material. As the viscosity of the filler-polymer mixture increases, it becomes difficult to process the composite. Many irregular shape fillers lead to a significant increase in the composite viscosity at higher filler content and ultimately lead to the wear and tear of the processing equipment. Wang et al. [81] artificially designed the fillers to study the influence of filler shape on the thermal conductivity. These fillers are equilateral triangle, spherical, square, rhombic, elliptical, rectangular, I-shaped, T-shaped, Y-shaped, double Y shaped fillers and quad Y shaped. The shape of the CB aggregate affects the bulk density of the product. In general, bulk density: spherical > elliptical > linear > ramified. The opposite is applied for electrical conductivity. They found that the filler with long heat conduction distance and having large contact areas are the best for thermal conductivity (double Y shaped fillers followed by Y shaped fillers). The orientation of CB fillers can be controlled during materials processing [80]. Chen et al. [80] mentioned in their review on thermal conductivity of polymer-based composites that the thermal conductivity of the plate-like fillers, aligned “in-plane” are larger as compared to “through-plane” or normally aligned fillers. Kanbur et al. [82] prepared PP/CB composites (varying CB weight percentage) through melt blending using Brabender. These test specimens were fabricated through compression and injection mouldings to study the effect of polymer chain orientation on the mechanical properties. It is observed that the mechanical strength of injection moulded samples is higher than the compression moulded ones [82]. In the LTW of polymers, CB plays a vital role as one of the compounding ingredients in the absorbing partner [83]. Due to the unequal distributions of CB or improper dispersions of CB, the absorption of the laser beam is unequal/preferential on the absorbing surface, which is in contact with the LT. Various hypotheses are formulated. One of them is a polymer matrix containing more CB that will absorb more laser energy. Due to the presence of CB, the melting point of the polymer is reached faster as it has greater heat-absorbing properties. This material surface melts first and in higher volume as compared to the polymer matrix having lesser CB content. Hence, there is an unequal melting of the surface resulting in an unequal weld width.

To increase the penetration depth and hence the weld strength, the carbon content needs to be decreased but with high laser power. However, high laser power can lead to degradation of the polymer [84]. I. Jones et al. [85] developed software that estimates the amount of absorber required for the given polymer material to be welded and also predicts the laser power and absorber dispensing equipment necessary for the application. Laser absorption and the heat flow characteristics of the polymer are utilised to make the necessary predictions which can reduce trial and error during experiments. It was shown that weld strength has some correlation with HAZ depth which makes HAZ depth an indicator of weld quality.

CB also affects the crystallisation of isotactic PP as a filler in the composite. Mucha et al. [86] studied the nature of nucleation and crystalline growth mechanism of PP which changes with the carbon content (Sakap 6, Carbochem, Gliwice) and the crystallisation temperature (T_c_). CB was varied in the amount of 2%, 5%, 10%, 20% and 30% weight fraction, which is mixed with a previously obtained sample containing 40% CB. In samples containing a small content of CB, it is observed that nucleation and crystallisation rates increase significantly. The slower/faster rate decides the morphology of the polymer matrix, which defines the properties of the two joining parts.

The polymer formulation and its processing, as well as the methodology of material production, affect the transmission properties. In this, the granules are either compounded with CB or it is added during the injection moulding in an extruder. Studies have shown that the laser light transmitted through the thermoplastics are a function of thickness [87] and the laser incidence angle. There is a decrease in the transmission rate with an increase in the laser angle of incidence [88]. Transmission of laser energy in polymer reduces due to an increase of crystallinity and large spherulites [89]. Apart from CB’s content, the size and distribution of the particles also impact the polymer properties. Compared to other fillers, CB is cheaper, and hence, it has gained industrial importance. However, there are certain products where black colouration is not preferred. Special pigmentations have been developed and studied by Haberstroh et al. [90] using a high power diode laser (maximum power 80 W) at a λ of 940 nm which are non-black (Clearweld^®^, Gentex Corporation, Cambridge, UK and Lumogen^®^, BASF AG, Ludwigshafen, Germany) and can absorb the laser beam. However, these non-black pigments were unable to weld micro parts and micro-structured parts. Further studies are needed for micro-welding parts containing non-black pigments.

#### 3.1.2. Carbon Fibres (CF)

Carbon fibre reinforced thermoplastics (CFRP) are widely used in aircraft and vehicles to meet their lightweight objectives. However, it is difficult to join thermoplastics (such as polyphenylene sulfide: PPS, an SC polymer) using adhesives as they are resistance to chemicals. An extensive surface preparation step is often involved for such polymers for adhesive bonding. Another technique called resistance welding usually involves an additional conductive element within the welding zone which could be undesirable for the finished product. In induction welding, the polymer matrix needs to be electrically conducting, and the weld zone and HAZ are to be conditioned by the coil geometry. LTW can be utilised to overcome these difficulties for welding CFRP products having stringent tolerance requirements. The type of absorption in the case of CFRP LTW is entirely different compared to unreinforced polymers. In the case of unreinforced polymers filled with CB, the heat source responsible for the heat conduction into the LT part is located at the surface of the LA part. However, in the case of CFRP, it is located inside the LA partner [26]. Jaeschke et al. [26] observed that the laser beam was completely absorbed in the CFs within the reinforced LA part after passing through the LT part. The heat conduction starts at the carbon fibres. The thermal conductivities of CFRP and unmodified polymer matrix is highly different. Hence, higher line energy is required in case of CFRP to ensure an equal maximum temperature distribution. Jaeschke et al. [26] studied the LTW of high-performance polymers and reinforced composites. With the help of a thermal camera, they observed a 360 °C temperature while welding unreinforced PPS with laser power of 4.0 W and scanning speed 1000 mm/s. For achieving the same welding temperature, the speed was reduced to 250 mm/s at constant laser power for the PPS containing CFs. It was also observed that the temperature profiles depends upon the arrangements/orientations of the fibres, i.e., complete transverse, complete longitudinal, half transverse, half longitudinal, centred polymer depot, etc. as shown in Table 2:

The fibre arrangement also defines the shape of the weld seam. In the same study, the weld seam was a spherical shape in the case of complete transverse fibre arrangement whereas, in case of centred PPS depot, the weld depth decreases resulting in an elliptical, flattened shape. Wippo et al. [91] also investigated the influence of the carbon fibre orientation on the weld seam orientation. For the welding process in the parallel direction to the CFs, the heat stayed within the weld seam area. For the perpendicular welding process, the heat conduction was outward from the weld seam along with the CFs. Due to this, the temperature in the middle of one scanning line was lower and higher on the weld seam edge. The perpendicular fibre orientation resulted in better homogeneous temperature distribution (measured by the infrared camera) compared to parallel direction. While studying the influence of CFs on the temperature distribution during the LTW process, Wippo et al. [92] observed higher weld width when welding PPS containing CF in the perpendicular direction than welding along with the fibre. However, this was more evident for higher line energy. At line energy = 1 J/mm, the weld seam was 1.6 times higher for 90° fibre orientation than 0°. They utilised a thermo camera and a pyrometer for monitoring the temperature. The heat conduction along the CF direction was higher than perpendicular. The lap shear strength of the welded samples with CF reinforcement is often higher compared to unreinforced LA polymer [26]. CF does provide good mechanical properties, but care should be taken to design its concentration and orientation as per the desired properties.

#### 3.1.3. Carbon Nanotubes (CNTs)

Carbon nanotubes (CNTs) (multi-walled CNTs (MWCNTs) and single-walled CNTs (SWCNTs)) are known for their exceptional thermal, mechanical and electrical properties, enhancing the physical properties of polymers when blended into the matrix. These are considered to be potential nanoparticulate functional fillers for polymers [93,94]. However, their dispersions are difficult due to their larger surface area and stronger interactions. By adding the compatibilising agents in the filler-polymer mixture, the dispersion can be enhanced. The functionalisation of the nanotubes can further improve the interactions. However, this method may interrupt the CNTs molecular backbone resulting in poor mechanical properties. Ultrasonication and melt mixings are also suitable methods for the dispersion of CNTs in the polymer matrix. Pan et al. [95] prepared the composites of untreated and hydrochloric acid-treated MWCNTs along with neat PP, and Maleic Anhydride grafted PP (PP-g-MA) by melt blending. This results in large and uneven dispersion of filler aggregates in the composites of untreated MWCNTs and a neat PP matrix compared to treated MWCNTs. Composites with acid-treated nanotubes and grafted PP result in smaller aggregate size with uniform dispersion among all methods. However, acid-treated MWCNTs show poor electrical conductivity. It is believed that the decrease in the aspect ratio (length to diameter ratio) of the nanotubes improves the dispersion but increases the distance between them that prevent the conductive network formation. Wu et al. [96] reported that for a better network, thermal conductivities of the composites with smaller particles and fillers are relevant. These studies highlight that to enhance the network structure of the CNTs in the polymer matrix, a balance between its dispersion and conductive network formation is essential [95]. Due to these properties, CNTs are added to the LA part of the joining component of LTW of polymers which is an unexplored area [94]. They can be welded in a similar way as CB based polymer composites [94]. Rodríguez-Vidal et al. [45] studied the welding capability as a function of the CNT concentration in the ABS (acrylonitrile/butadiene/styrene) sheets using a diode laser system (maximum output power: 50 W, λ: 808 ± 4 nm). With an increase in the CNT content, the laser energy requirement decreases for the same welding seam. In the case of the ABS studies, the percentage of CNTs was from 0.01% to 0.05% by weight. By adding just a certain amount of CNT fillers, the mechanical properties of ABS were improved. At higher dosage of CNT concentration, the ABS composite became more sensitive to the variations in input power and at a lower dosage of CNTs, scanning speed was more relevant. They observed that at λ 808 nm, the absorption coefficient of ABS composites with higher CNT dosage was almost 1.6 times higher than that for lower CNTs. Hence, higher power values were required for a lower dosage. At higher concentration of CNTs, the laser penetration depth decreases resulting in thinner layer thickness due to limited physical and chemical mixing. The thinner layer will not be able to maintain a high shear stress of the welded joint. Generally for semi-crystalline polymers such as ultra high molecular weight polyethylene (UHMPE), 0.1–0.5% of CNT is an optimised dosage range as observed by Visco et al. [97] during welding UHMPE-CNT composites using 3-ns Nd: Yag laser (532 nm). The surface damage such as small cavities and material detachment in 0.1 wt.% CNT composites were less visible as compared to samples containing 1.0 wt.% and 10 wt.%. Torrisi et al. [98] used the same laser system with various filler for UH composites. They found that the best polymer coupling was with CNT which showed a good adhesion value with 10 MPa shear rupture at irradiation time of only 2 min.

#### 3.1.4. Emerging Fillers in LTW

Apart from the CB, CNTs and CFRP, various other fillers with higher thermal conductivities are used in the LA layer, which is not fully explored. Boron Nitride (BN) has a thermal conductivity of 250–300 W/mK, and that of CB varies in the range of 6–174 W/mK at 25 °C due to which there is an emphasis on using BN as filler in composites for various applications [99]. BN is not used widely in LTW process as per our knowledge. However, the literature on the interaction of BN with the polymer matrix and its effect on the thermal and mechanical properties can form a basis for its application in LTW of SC polymers. During the investigation of thermally conductive polymers, Kovacs et al. [100] studied the effect of the processing method and the effect of hexagonal BN on the thermal conductivity of PP. The samples with 30 vol. % BN was found to have four times more thermal conductivity as compared to pure PP. The studies also helped in understanding the effect of the moulding process and conditions on thermal conduction. They found that the compression-moulded samples possessed higher thermal conductivity than the injection-moulded samples because of their shell-core structures. Due to the lower content of the filler on the skin layer, the heat transfer decreases for injection-moulded samples. The thermal conductivity of the compression-moulded samples is found to be 60% greater than the injection-moulded samples. The melt flow index (MFI) of the composites decreases with CB content due to the adsorption of polymer molecules on the CB surface [101]. The percentage of crystallinity decreases with an increase in CB content due to deformation in the crystalline structure. Although the thermal stability of the material increases with the CB content (>35 vol.%) [102], Chen et al. [80] identified that the material became more brittle after a certain weight percentage of CB. Similar studies on the effect of BN on the MFI and percentage crystallinity of SC polymers could further facilitate BN use in the laser field.

The use of the hybrid fillers in polymer composite is another technique which facilitates the amalgamation of two or more filler properties in a composite. Fabrications of hybrid filler systems with different shapes, sizes, and filler types ensure physical properties’ tunability [103]. These also maximise the filler packaging density and improves the viscosity of the system. In the study of thermally conductive polymer, Kovacs et al. [100] reported that talc, along with hexagonal-BN results in an improved thermal conductivity as a result of synergetic behaviour. Choi et al. [104] also mixed inorganic fillers of different types and sizes in the polymer composites resulting in high thermal conductivity. Figure 5 shows the hybrid filler composite system (large-sized and small-sized) of AIN and Al_2_O_3_ studied by Choi et al. [104]. This technique also helps to overcome void formation in the matrix and leads to better thermal conducting paths. Each conducting path is composed of the CB conductive particles. The conductive paths are formed depending upon the dispersion and distribution of CB in the polymer matrix.

Carbon nanofibres (CNFs) can also be used in SC polymers which can absorb the infra-red region of light for the diode laser welding process. This material has a broader scope in joining insulating and conductive materials in the field of chemical and biological sensors to fuel cells. In general, the size of CNFs is larger than CNTs (1–10 nm in diameter) but smaller than CFs (0.005–0.010 mm in diameter) Dosser et al. [94] used integra direct-diode laser of λ 810 nm with a maximum power output of 60 W for welding poly-ether-ether-ketone (PEEK) containing CNFs and CB. For PEEK-CNFs composites, the peak weld strength was 402 psi while that for PEEK-CB composites was 302 psi. A higher average power was required for CB composites due to the low weight percentage of carbon material.

Similarly, CB can be used with other fillers to enhance thermal conductivity. Kang et al. [105] used 50 wt.% Al and MWCNTs hybrid filled PP composite, which leads to an enhanced thermal conductivity of 0.7 W/mK. The Al-CNTs were prepared using a co-rotating inter-meshing twin-screw extruder and ball-mixing of PP with Al-CNT, carried out using different ball sizes (1.0 and 2.0 mm) for 6 h. Figure 6 shows the schematic diagrams for the dispersion of CNTs into the PP matrix. However, the embedded CNTs dissociate apart from Al due to high shear during compounding. These CNTs play a significant role in bridging Al-CNTs in the PP matrix resulting in the improvement of thermal and mechanical properties of the PP composites.

In the LTW of polymers, CB plays an essential role as one of the compounding ingredients in the absorbing component [106]. If the dispersion and distribution of CB in the polymer matrix is homogeneous, then there will be uniform heat distribution resulting in better weld properties. However, weld integrity depends upon various factors such as polymer matrix, filler type, laser parameters [107], etc. The diode laser can penetrate a few mm for most of the non-pigmented and unfilled thermoplastics. Various compounding ingredients such as fillers, pigments, processing aids, etc. affect polymer morphology [83]. Haberstroh et al. [83] investigated by mixing CB between 0.1 wt.% of 20 nm and 0.5 wt.% of 60 nm to the base material in an extrusion process. The dimensions of the aggregates of CB particles found to affect the laser light absorption when the λ is ~940 nm. More extensive heat transformation is observed with smaller particle sizes by keeping the irradiated laser power constant, which yields higher weld strengths. At a given energy input, there is an increase in temperature of the welding surfaces with the CB content and an increased flow velocity and finally higher weld strength is achieved.

Jansson et al. [108], determined the welding parameters using the quasi-simultaneous laser welding technique for PP by varying CB and fibreglass contents using diode laser at 940 ± 10 nm and focal length 163 mm (focal spot: 1 mm). They found that by increasing the carbon content of the absorbing partner from 0.05 wt. % to 1.5 wt.%, an enhancement in the weld strength up to a specific limit (22–24 N/mm) can be achieved. With 0.1 wt. % of CB, the weld is found to be asymmetrical, whereas 0.5 wt. % of CB results in a symmetrical weld as observed in the micrographs. A higher CB content results in better heat transfer to the transmissive part of the polymer due to the change from volume to surface absorption. These observations are consistent with Acherjee et al. [109] who studied the effects of CB on temperature and weld profiles in LTW of polymers using a transient numerical model based on conduction mode heat transfer. Natural and opaque polycarbonate (PC) was used in the experiment. The optical penetration depth of opaque PC was 63 μm at the λ of 808 nm. They observed an increase in the weld width with an increase in the CB content in the absorbing polymer. In the surface absorption, the radiant energy is completely absorbed in a very thin layer of the absorbing portion. At the surface, the absorbed energy gets converted into heat and results in a wider weld width. However, if the CB content is high, the laser energy is absorbed within a thin surface layer due to low penetration [89].

### 3.2. Effect of Pigments/Nucleating Agents in the Laser Transmissive Layer

When the laser passes through the laser transmissive layer of the polymer, as illustrated in Figure 7, some of its energy is lost due to absorption and/or scattering by the compounding ingredients like fillers, pigments and crystal structure. In general, when any source of electromagnetic radiation passes through an object, three phenomena occur transmittance, reflectance, and absorbance [110]. The percentage of each of these depends upon the morphology of the specimen. If the radiation travels in a straight line through the polymer, then the polymer is said to be “non-scattering” (Figure 7a); otherwise, it is referred to as “light-scattering” that may be partial or complete (see Figure 7b,c). In a multi-scattering polymer (like SC polymers), some percentage of light energy can pass through the interstices while the rest gets scattered. For an optically transparent material, a low crystallinity with a small crystallite phase is required. The additives are added to enhance the mechanical and thermal requirements of the final product [111]; however, the infrared wavelength (700–1200 nm) can still pass through the laser transmissive layer.

For many polymers applications, it is desirable to have good transparency or decreased crystallinity [112]. This can be done by copolymerisation of polymers such as PP with small quantities of ethylene, 1-butene, 1-pentene, or 1-hexene. Copolymerisation has better flexibility and transparency as compared to homo-polymers.

Isotactic polypropylene (iPP) has a complex and multifaceted crystalline architecture [113]. The α-structure of PP is the most widely occurring crystal structure, which is a monoclinic lattice. The hexagonal β-structure is a minor constituent of the bulk sample. The third polymorph is a triclinic lattice, γ-structure. The addition of nucleating agents to SC polymers controls the crystallisation process of iPP [114] as they provide a large number of nuclei. Nucleating agent modifies the crystallisation behaviour of iPP by accelerating the formation of crystalline structure, decreasing the spherulites dimensions and promoting crystallinity which improves thermal and mechanical properties [111,115,116]. Kersch et al. [111] studied the influence of two 1,3,5-benzenetrisamide beta-nucleating agents on the morphological and mechanical properties of iPP. They found a remarkable increase of impact strength (138%) or an improved toughness for the materials containing benzenetrisamide beta-nucleating agents. Macauley et al. [117] found that by varying the crystallisation conditions, crystals of varying structures and morphologies can be generated and to a certain extent, nucleating agent controls the ageing process [117,118,119,120]. Feng et al. [121] added dibenzylidene sorbitol (DBS) to the iPP to study the effect of nucleating agent addition on the iPP crystallisation. They found that the rate of crystallisation increases with the DBS amount. It is indicated that iPP results mostly with truncated 3-dimensional spherulites, whereas sporadic nucleation occurs at lower T_c_. The addition of DBS results in a spherulitic morphological change. The amount of laser beam transmitted, absorbed, scattered or reflected depend upon the matrix morphology. An increase in crystallinity enhances the mechanical properties of the polymers to some extent. However, for an enhanced laser transmission, a lower percentage of crystallinity is recommended. It is challenging to determine an adequate dosage of the nucleating agent as one of the compounding ingredients in the LT layer which could provide a certain crystallinity for the mechanical strength of the final product as well as optically transparent to the desired laser beam used for welding.

Qamer Zia et al. [122] found that light transmission increases when the size of the spherulites decreases. Light transmission exceeds 90% in films of 100 µm thickness in the absence of spherulites as observed through the ultraviolet-visible (UV-vis) spectroscopy. For low-see through clarity samples, the crystallinity is 60–70%, with an average spherulite size of about 50 μm. During the LTW of PP using a diode laser, Ghorbel et al. [49] carried out the geometrical and microstructure characterisations of the weld and observed very well-distinguished zones in the welded sample using Microscopic Fourier transform infrared spectroscopy (MFTIR): zone 1, zone 2 and zone 3 (see Figure 4). In zone 1, due to the high melting temperatures of PP, there was the total removal of initial spherulites. Only a limited number of nuclei grow (low nucleation density), leading to the formation of large spherulites of 50 µm in size. Zone 2 was an intermediate size of spherulites which were larger than the initial PP but smaller than zone 1 due to limited melting duration. Zone 3 contains spherulites similar to initial PP to a large extent as the temperature in this region is around 10 °C below the melting point. Some of the new crystals that were formed in zone-3 offered sufficient crystal surface for the melted polymer to recrystallise on it upon cooling (secondary nucleation or spherulite growth). It was observed that smaller spherulites in the weld result in more uniform mechanical stress distribution. However, the influence of the microstructure and crystallinity rate onto the mechanical properties of the laser welded samples are still under investigation for the given composite of PP. Pelsmaeker et al. [123] carried out clear to clear laser welding for joining thermoplastic polymers [123] using a Thulium fibre laser at λ 1940 nm (FWHM linewidth: 0.7 nm). They suggested that the welding performance can be compromised due to the scattering of the incident laser light. According to the literature [124], very fine spherulitic microstructures were produced while adding some nucleating agent at a certain dosage. Figure 8 shows the refinement of spherulites induced by increasing levels of sodium benzoate in iPP [125]. Due to the fine microstructures, there was an enhancement in the yield strength, ductility, and impact strength of the polymer matrix. Studies have shown that foreign particle nucleation results in improved ductility and strength [124].

## 4. Effect of Polymer Processing Parameters on the Properties of Semi-Crystalline Polymer

Materials scientists and engineers have recognised a relationship between fabrication processing and the internal morphology of the materials [126]. Besides, the compounding ingredients, different processing techniques and parameters result in various types of end products with varying properties (process–structure–property relations) [127]. Various types of microstructures result in different morphology and ultimately material properties in the case of laser-welded SC polymers. When the laser beam is emitted from the laser and reaches the polymer, it undergoes recrystallisation for the laser generating the continuous wave. The morphology developed after recrystallisation depends upon its initial morphological structure formed during polymer processing. Hence, it becomes essential to understand changes taking place in SC polymers due to thermal history, spherulite formation, etc., during the fabrication process. Processing techniques such as extrusion and moulding involve crystallisation during the operation when the polymer undergoes shear stress of the crystallising volume. It is essential to understand the role of this shear stress in controlling nucleation. This helps relate the spherulitic structure or polymer morphology to the various types of structures found in industrial products [128].

The selection of the type of polymer also plays an important role. PP random copolymer (PPCP, often copolymerised with ethylene) is a type of SC polymer. It has high impact strength and flexibility with better clarity as compared to PP homo-polymer [129]. Stereoregular blocks of shorter lengths are formed due to the random addition of ethylene repeat units along the backbone of PP. These ethylene units hinder the crystalline arrangement and thus leads to low melting temperature and crystallinity [118,130]. PPCP is widely used for injection moulding applications, especially where high clarity is required [62]. Random copolymers have a lower heat distortion temperature, less rigidity, but higher impact strength. Material toughness increases with an increase in ethylene content. Due to this property, PPCP allows higher laser transmissivity when used as LT. Homo-polymers of PP is brittle for temperatures higher than 0 °C for low MFI polymers [131]. The brittleness temperature for the homopolymer of MFI 3 is around +15 °C whereas it is 0 °C for the homopolymer of MFI 0.2. In the case of a copolymer of MFI 3, brittleness temperature is around −15 °C and that for MFI 0.2, it is around −20 °C [131]. As the crystallinity increases, surface hardness, and scratch resistance increases. During LTW of amorphous and SC PEEK for medical application, it was found that the bond strength of SC PEEK is higher than the amorphous PEEK bonds for all the values of power and speeds [10]. Higher strength is due to the packing morphology of SC polymers.

MFI of the polymer is to be considered before one sets the moulding or extrusion parameters. It measures the molecular weight distribution. MFI helps to determine the temperature and clamping force required in the process to maintain consistent product quality, without creating waste material. The molecular weight of PP is one of the parameters which define the ability of the polymer to fill the mould. MFI indirectly indicates the uniformity of polymer properties to some extent. The flow rate is an empirically determined parameter influenced by the molecular structure and physical properties of the polymer [132]. If the polymer has low flow resistance upon application of heat, it will facilitate the inter-diffusion phenomenon during LTW process.

In injection moulding, there is an inter-relationship between the temperature and pressure on the various transitions. During the filling of the polymer melt inside the sprues, runners, and gate of the injection moulding machine, there is no pressure inside the mould. Gradually, the mould gets filled up and generates pressure inside. Clark et al. [128] mentioned that the initial polymer that enters the mould undergoes rapid crystallisation while the additional polymer spurts under high pressure inside the mould. The core and the material near the mould cavity will have different microstructures. Besides, due to the rapid increase in the melt pressure during cavity filling in the injection stage, flow lines are formed in injection moulded samples, which weakens certain areas. Hence, care should be taken while selecting the samples for laser welding trials. In polymer melt, compressibility should also be considered along with the effects on morphology. The macromolecules present in the melt gets extended and orientation occurs due to the external flow which is known as orientation-induced crystallisation. Under the quiescent condition, sheared polymer melt will have different crystallisation behaviour compared to melt crystallisation. The crystallisation of iPP quiescent melt occurs slower compared to its sheared melt [133]. The sheared melt iPP leads to the formation of β spherulites. These polymers when laser welded will have higher impact resistance.

A twin-screw extruder (TSE), either co-rotating or counter-rotating, is widely used to process PP [134]. In the case of single-screw extruders, polymer experiences a wide range of mixing history depending upon its location within the extruder. Whereas, there is a uniform shear distribution within the polymer extruded using a TSE. For lab testing purposes, melt compounding of PPCP can be done in a co-rotating micro-compounder. Verma et al. [62] used the HAAKE MiniLab II micro-compounder to prepare nanocomposites with varying CNTs followed by microinjection moulding for test sample preparation. In their research, the melt blending of PPCP was done with CNTs at a processing temperature of about 200 °C with a mixing time of 10 min and a screw speed of about 50 rpm [62]. To obtain a uniform dispersion of CNTs, Sven Pegel et al. [135] carried out the mixing of MWCNTs at higher temperatures of around 250 °C. However, at 300 °C, poor dispersion is observed when compared to the dispersion obtained at 250 °C. One of the reasons could be higher viscosity at 250 °C that facilitates dispersion. For making PP clay nanocomposites, Treece et al. [136] compared the ability of melt blending in single screw and TSE. It was concluded that TSE is more effective due to more shear. The L/D ratio (flighted length of the screw to its outer diameter) of TSE is 52 with D = 27 mm, keeping the barrel temperature from 200–210 °C with co-rotating screws (200 rpm). In the case of poor dispersion, the samples will not weld uniformly during LTW process which will lead to points of stress development.

Lertwimolnun et al. [137] studied the effect of processing conditions on the PP-organoclay nanocomposites in a co-rotating TSE. L/D ratio of the screw was 24 with a screw diameter of 50 mm and keeping barrel temperature around 180 °C. They found that processing parameters do not show any effect on the state of intercalation, which is interpreted by interlayer spacing. The level of exfoliation increases with a decrease in the feed rates and an increase in the screw speed. Uniform dispersion and distribution of the compounding ingredients are the fundamental requirements in achieving an optimum material property for any product [138].

The cooling rate of the polymer melt affects crystallisation [118]. Mileva et al. [118] found that non-isothermal melt crystallisation at slow rate results in the formation of monoclinic α-crystals with lamellar geometry. Lamellae thickness and degree of crystallinity decrease due to the addition of comonomers like ethylene and affect mechanical, thermal and optical properties [139,140,141,142]. Mileva et al. [118] used isotactic copolymers of propylene with ethylene and 1-butene with different cooling rates. This results in the formation of α-crystals due to ethylene and 1-butene co-units that coexisted with a small number of γ-crystals. The degree of crystallinity and its rate decreases due to the exclusion of ethylene co-units as compared to 1-butene units. Various structures of the crystalline phase of PP have already been mentioned in the previous section. The monoclinic α-structures are formed on melt-crystallisation without any special nucleating agent [143]. In the presence of the nucleating agent, heterogeneous β-nucleators, trigonal β-formation occurs, and γ-polymorph, orthorhombic, are formed at elevated pressure during crystallisation [143]. In the case of the LTW process, there is a formation of different morphology due to the clamping pressure. The β-form is known for its mechanical performance. Impact strength, elongation at break and toughness of β-form is higher as compared to α-form [144]. The α-forms are known for their rigidity and low deformability. Rapid cooling of the polymer melts results in low crystallinity with a cooling temperature being less than the T_g_. Te polymers with low crystallinity can enhance the transmissivity of LT. However, a balance between percentage transmission and optimum mechanical properties for welded samples should be taken into consideration. Some of the aspects of the orientation of CB and the difference between injection and compression moulded products are already discussed in the previous sections. There is no direct correlation in the literature for the effect of polymer processing with the LTW process. The clamping force applied during the LTW can be considered similar to compression moulding process. Shangguan et al. [145] observed that during compression moulding of the molten iPP, β–crystals were formed when they were allowed to crystallise at natural cooling rate. The content of the β-crystals was increased with an increase in the melt compression pressure. The same theory could be applied by increasing the clamping force during LTW process.

## 5. Melting and Inter-Diffusion Phenomenon of Semi-Crystalline Polymers

### 5.1. Polymer Melting and Solidification

In the process of LTW of polymers, the laser beam is absorbed by one of the joining parts, and melting occurs within the material. Hence, understanding the melting and inter-diffusion phenomenon is a vital consideration for laser welding. The melting of the polymer, followed by its solidification, may lead to various morphological changes. Like PP, SC thermoplastics consists of crystalline and amorphous regions [146,147,148,149,150,151]. They have a glass transition temperature (T_g_) in the amorphous regions and crystalline melting temperature (T_m_) in the crystalline regions. The crystalline melting point is higher than the T_g_. These polymeric materials flow above T_m_, where no crystalline regions are present [152]. Crystal thickness and topological constraints in the amorphous region define the melting of the crystalline component in SC polymers [153]. The chain mobility of the polymer increases above T_m_ or T_g_ in the weld region. This allows the chains to diffuse across the joint interface and get entangled along with the chains of another side of the interface. The joint strength and weld formation between the surfaces occur by this mechanism, which is explained in detail. The network path of the CB, destroyed during high shearing, can be repaired during melting, followed by re-solidification of the polymer composites.

SC thermoplastic resins consist of crystalline as well as amorphous domains, i.e., they have imperfections in the crystallites and non-uniformity in the crystal sizes. On a technical scale, complete crystallisation of plastics is not feasible due to monomers’ statistical distribution and the polymer chain structure. The loop formation of polymer chains may hinder the chain structure regularity, chain mobility, and molecular weight of plastics. Certain polymer chains may be partly in the amorphous phase and partly in the crystalline lamella. The chains which start in one lamella, crossing the amorphous zone and join another lamella, are called tie chains. These act like polymer crosslinks, providing better mechanical properties to the polymer matrix of SC as compared to amorphous polymers. During the polymer melting process with laser welding, if all chain strands entered the melt, there would be extreme pressure generation near the crystal surface due to an increase in the density [154]. As a result, mechanical stress gets concentrated at increasingly fewer sites on the surfaces of the crystallites.

Partial alignment of the molecular chains occurs during polymer crystallization. The folded chain in an arranged form is known as lamellae or crystallites comprised of larger spheroidal structures known as spherulites. Crystallite size and proportion (α-, β- and γ-crystal modification content) determine the value of T_m_ of SC thermoplastics [155]. The higher the size and the proportion, the higher is the T_m_ value. There is approximately 10 °C difference in the melting points of α- and β- form of crystals. The melting of the α-form lamellae is above 164 °C while that of the β-form lamellae is near 152–164 °C [156]. The determination of the thermal properties by calorimetric methods provides information on the motion possibilities of the polymer chains, heat capacity, and crystallinity.

Melting behaviour influences the processing and end-use application of SC polymers like PP [157,158]. These polymers are said to be more ordered as compared to amorphous polymers [159]. The long-chain molecules are packed closely in a regular manner. However, all polymers are amorphous during the melting process. The melting behaviour of polymers showing polymorphism is complex, such as in the case of PP [158,160]. PP has mainly three crystal modifications, namely, monoclinic (α), hexagonal (β) and triclinic (γ) with α form being the most stable [160,161]. As mentioned previously, there is a melting temperature difference between these crystal structures. These crystals will be present in different weight proportion in the polymer matrix. The laser parameters need to be adjusted depending upon the melting stability of the polymer. Higher laser power values or lower scanning speed are required for PP with a higher percentage of stable crystalline structure.

In the temperature range of 0–140 °C, Ullmann et al. [162] observed only monoclinic α–modification of iPP. PP contains a nucleating agent and shows α– and β–modifications from 0–130 °C. The β–modifications are metastable compared to the α–modification. Zhou et al. [158] proposed three mechanisms to understand the morphological changes during polymer melting. The first mechanism is about surface melting, where all lamellar crystals melt simultaneously from the surface for an extended period. According to the second mechanism, there is a sequential melting of individual lamellar crystals within a polymer stack depending upon the stability. The thinner crystals melt first while the thicker crystals are not affected. The third mechanism is based on stack melting, which states that the thinner or less stable crystals of the crystalline lamellar stacks melt entirely, retaining the internal structure of the stacks. Many researchers agree on the lamellar thickening during the melting process of SC polymers [163,164,165,166]. In the LTW process, the melting of the material should be uniform to avoid any product defects. For example, if the second mechanism is followed, the thinner and thicker crystals have to be uniformly distributed across the polymer matrix during the melting process. LTW is a melt-recrystallisation process that can be studied using temperature-modulated differential scanning calorimetry (TMDSC) as a characterisation technique.

The processing conditions of polymers also influence crystallinity. An increased crystallinity can be obtained by slow cooling of the polymer melt or tempering at Tc. It is observed that annealing of the SC polymers below their melting temperatures leads to an increase in the crystallinity and crystallite thickness/lamellae thickness [125]. There are three temperature steps during the annealing process. In the first step, there is a decrease in the density [125] followed by a gradual increase in crystallinity occurs in room temperature cooling [167]. On the other hand, crystallisation gets hindered by the quick cooling of the plastic melt. The effect of holding a polymer sample at a high temperature followed by quenching is similar to holding a polymer sample under mechanical tension and then releasing it [125].

The cooling rate affects the morphology and crystal structure of the polymer melt. So far, it was believed that the melting temperature of α-iPP spherulites grown at low supercooling is higher than that grown at high supercooling. However, in the study by Zhou et al. [158], an abnormal melting behaviour is observed in which melting temperature spherulites grown at high supercooling is higher than that of low supercooling. Crystals grown at very low supercooling have higher perfection and consequently, higher melting temperature. The laser welding parameters will differ for crystals grown at different supercooling conditions. Piccarolo [168] observed only α-monoclinic phase below the cooling rate of 20 °C/s and above this value, there is a coexistence of mesomorphic and α-monoclinic phases. Virgin PP is initially transparent and it becomes gradually translucent as the melt cools to room temperature. The chains of the polymer rearrange themselves from the amorphous phase during the melt to an ordered SC phase in the solid sample. The solidification of the SC polymers occurs over a narrow range of temperatures. The range depends upon the type of polymer and its compounding ingredients. The cooling of SC polymer melts results in a more organised structure as compared to amorphous polymer melt.

In the LTW of polymers, the injection or the compression moulded product undergoes re-heating, followed by re-solidifications. The crystal structures formed during the fabrication process undergo a regrowth phenomenon [126]. Bryce Maxwell [126] found that upon re-cooling, the nucleation and spherulitic growth occur in the periphery of the old spherulites. During the initial growth process of the spherulites, there is an occurrence of dendritic segregation of the impurities from the spherulite centre towards the periphery. The foreign matter deposited in the periphery of the old spherulites acted as a nucleation site for the new crystallisation upon re-cooling. The polymer’s thermal history is one of the essential parameters in the control of the morphology during LTW. A fine “grain structure” can be formed instead of large, well-formed spherulites during re-melting under proper conditions. An understanding of the material thermal history can be beneficial for establishing laser processing conditions that can enhance weld integrity.

### 5.2. Shrinkage

The shrinkage percentage of SC polymers is high as compared to the amorphous polymers. This is due to the packing of the polymer chains in an ordered manner as per the moulding parameters. Slow cooling rates allow polymer chains to disentangle themselves and form their crystalline structure resulting in higher crystallinity. This results in superior mechanical strength; however, the shrinkage percentage is higher. Rapid cooling of the melt inhibits crystallisation, resulting in a polymer with lower crystallinity, leading to poor mechanical properties compared to higher crystallinity. The shrinkage percentage is lower, but there are occurrences of dimensional instability in the later stage of the process known as after moulding shrinkage. In SC polymers, there is a gradual shrinking of the material in the direction of tension after its release from tensile stress. LTW is a type of heat treatment for polymers. There are chances of the polymer composites to undergo shrinkage upon cooling from their melt stage after laser welding. The shrinkage phenomenon is not emphasised much in the LTW field so far. However, it is a factor that should be taken into consideration for better weld quality. It was observed that as the thermoplastic cools down, it contracts perpendicular to the weld.

### 5.3. Inter-Diffusion Phenomenon

In the case of laser welding, inter-diffusion between the two surfaces occurs. Wu et al. [169] reviewed the inter-diffusion of polymer surfaces. They stated that when two surfaces of the polymer come into contact at a temperature above its T_g_, relaxation of the chain conformations at the interface occurs towards those chains in bulk. This is because of the Brownian motion. Gradually, the interface of the assembly disappears by healing the voids at the contact area. This leads to the development of mechanical strength [169]. Micro-Brownian motion leads to healing of the polymer chains and local flow, leading to good interfacial joining and, ultimately, restoration of the original surface contours.

Wool and O’Connor [170,171] used a five-stage model to explain the adhesion mechanism:(i)Surface rearrangement(ii)Surface approach(iii)Wetting(iv)Diffusion(v)Randomisation

Figure 9 shows the phenomenon of healing stages via inter-diffusion of the two polymer surfaces A and B. According to Wool and O’ Connor [170], it is a complex phenomenon and explained through various functions such as healing function, wetting distribution function and intrinsic healing function. During healing, the diffusion stage is considered an essential step controlling the development of mechanical properties. Complete loss of memory of the crack interface occurs during the randomisation stage. Through various equations, they could relate strength, elongation to break, fracture and impact energy parameters as a function of temperature, pressure, molecular weight, time and processing conditions.

The potential barriers in the interface in terms of inhomogeneity disappear at the end of the diffusion stage. The chains have more degrees of freedom in the stages of diffusion and randomisation. The strength of the polymer material appears during the last two stages [172].

In the initial stage, the chains are separated from each other by the interface. At t ≥ Tr, (tube renewal time: the amount of time required for a polymer chain to disengage itself from the initial tube completely) there is interpenetration and entanglement of the chains to the other side. This is explained based on the reputation model by Kim and Wool [171]. Figure 10 shows the chain conformations at various time intervals. At the initial stage of t = 0, the chain and its initial tube are at the interface. In the later stage, when t = t_1_, the minor chains started to escape from the tube and crossed the interface. At t = t_2_, the longer minor chains cross the interface and penetrate the other side. However, some small chain portions are confined in the initial tube. The diffusion and randomisation stages can be further characterised by considering the initial non-Gaussian to the final Gaussian conformation of the polymer chains (random walk model).

When two similar or dissimilar surfaces of the polymers are joined during the laser welding process, many factors are to be considered concerning the material. The difference in the melting temperature of the two materials should not be too high. Besides, if the viscosity difference is too high, the inter-diffusion phenomenon of the melts will not occur [173]. The melting temperature and viscosity of the LA changes with the CB content. Hence, a higher percentage of CB (depending upon its type) is not preferable in the LA. Above all, to allow the inter-diffusion of two joining partners, these should be chemically compatible with each other [2,174] to avoid the generation of harmful by-products. One study states that two dissimilar polymers are weldable if their equilibrium interpenetration depth is compatible with the polymer network mesh size [60].

It is preferable for laser welding to have one material with high transmittance of laser and the other with high absorbance [173]. To increase the absorbance of one of the joining partners, additives are added as a part of their formulations as discussed in the previous section [2].

Hopmann and Weber [2] developed a new concept for the melting of dissimilar plastic parts. According to their study, by inserting an absorbing film or a layer injected in the two-component injection moulding process, even unpigmented layers can be joined [2]. The laser beam is allowed to transmit through the laser-transmissive material. When the laser beam is absorbed by the second joining surface of the material, laser energy gets converted to heat energy. The laser transmissive material gets plasticised by the heat conduction phenomenon taking place in the joining zone region. The viscosity of the material decreases due to the temperature rise, and the layer in contact begins to join. As the contact is improved, the volumetric expansion of the material occurs due to internal joining pressure. This expansion leads to the confluence of the polymer melts. The mechanical strength of the joint begins to develop after the completion of the inter-diffusion of the two surfaces. The two surfaces to be welded are kept in contact throughout the entire process [1].

Processes such as plasma treatment of the joining surfaces are one of the technological and economic approaches for the welding of two surfaces, specifically for incompatible thermoplastics. In this technique, functional groups are incorporated into the polymer surfaces, followed by the cross-linking reactions [175,176]. Such types of surface treatments can be utilised to obtain stronger samples before laser welding.

## 6. Effect of Laser Parameters on the Semi-Crystalline Polymers

For laser transmission of thermoplastics, there are mainly three sets of properties that are essential for modelling and understanding the processes [177]:a)Optical properties:
laser transmission energy (T);laser absorption energy (A).b)Thermoplastic formulation, its dispersion, and distribution in the matrix which also defines its flow behaviour and determines the mechanical properties:
dosages of the compounding ingredients in mass% (reinforcing agents, heat stabilisers, impact modifiers, etc.);shape and size of the additives;type and content of the pigments;c)Thermal properties and type of microstructure (morphological properties).

Most of the above mentioned material parameters are already discussed in previous sections. In this section, our discussion will be limited to the effect of laser parameters on material properties. Laser power, scanning speed, laser beam size on the material surface and the clamping pressure are the most critical parameters that affect these materials [25]. Hence, particular diligence and broad experience are needed in determining the crucial specifications for a suitable laser [178].

### 6.1. Laser Power

Laser power is one of the most critical parameters in the laser welding process [179]. Increasing laser power is one of the ways to make production faster. Optimum laser power can enhance the weld strength of the final product [180]. However, this value depends upon other laser welding parameters such as scanning speed and repetition rates. Hence, researchers have attempted to understand the role of each parameter by keeping all other factors constant. Based on a large number of experimental runs, an optimum value for these parameters can be achieved [39,181]. It was observed that the contribution of laser power in the laser welding process is found to be 58% (statistically more significant than other parameters), followed by focal distance (31%) and scanning speed (11%) [25]. In the selection of optimum parameters using natural and opaque acrylics (0.1% CB by wt.) with a continuous diode laser operating at 809.4 nm, Acherjee et al. [25] found that the joint strength of 4 mm thickness material increases with laser power. By increasing the laser power, the heat input at the weld zone increases, and hence the amount of material melted would be more in quantity with an enhanced weld strength. They also varied the laser power from 19 to 28 W and above 28 W, they found a decomposition of the base material. However, the scanning speed needs to be considered as these parameters are inter-related with the final properties of the weld. With the laser power of 28 W, scanning speed 5 mm/s and focal distance of 9 mm, they achieved maximum weld strength. With increasing the laser power, the depth of penetration and weld width increases [11,23,182,183]. The thickness of the material determines the threshold power for welding. To understand the effect of laser powers on the depth of welding penetration using finite element method (FEM) of simulation, Casalino et al. [184] considered butt weld geometry for the penetration welding of the two cylinders using a system of 800 nm λ and average power of 30 W. They used two different powers, 2.5 W and 0.67 W, for welding and found the penetration depths ~2.17 mm and 0.59 mm, respectively. Laser power is directly proportional to penetration depth and weld width, as shown in Equation (5) [184]:(5)$$P=\;\frac{\omega RWa\rho \left(cT_{m}+\delta H_{m}\right)}{0.484}$$
where ω: rotational speed (rad/s)

*R:* radius of the rotational trajectory (m)

*W:* weld width (m)

*a:* penetration depth (m)

ρ*:* polymer density (kg/m^3^)

*T_m_:* melting temperature (K)

*c:* specific heat capacity (J/(kg·K))

$\delta H_{m}$: latent heat of melting (J/kg)

Based on the simulation model for temperature fields, Thomas Frick [185] observed that the polymer does not adhere well at low laser power. For higher lasers powers the temperature of the material increases leading to local thermal damage of the material in the welding zone. For any welding mode, it is essential to keep the temperature lower than the polymer decomposition temperature. If the operating temperature exceeds decomposition temperature, the weld seam will result in unstable size and ultimately decrease in the weld shear strength.

Line energy (J/m) is also a parameter related to laser power (Equation (6)). Mathematically [186],
(6)$$E_{line}=\;\frac{P}{v_{scan}}$$
where *P* is the laser power (W), and $v_{scan}$ is the laser scanning speed (m/s). 

To visualise the process behaviour and to determine the location of the process window, Russek [187] obtained a characteristic curve of strength Vs the line energy (Figure 11). At lower line energy (below point 2 in Figure 11), there is light adhesion. Point 2 indicates an optimum value for welding beyond which, materials get decomposed. Line energy does not consider the critical process influencing aspects such as intensity distribution, process time scales and beam shape. Hence, the comparison of the process is limited.

Laser parameter is decided based on the type of the polymer, its compounding ingredients and the morphology. Apart from material properties, beam diameter at the workpiece and scanning speed are also key determinants of laser power. The effect of CB (particle size, particle shape, CB structure, etc.) on various properties of the polymer, especially thermal properties, has been previously discussed. In general, increasing the CB content in the laser absorbing part increases the melting ability of the polymer, which can be measured through MFI. However, it depends upon the type of polymer and CB involved. Laser power or laser energy requirement may differ depending upon the type of filler. Polymer sample containing CNTs have higher thermal conductivity and hence the dosage required is low. Visco et al. [97] confirmed that the amount of CNT filler (for ultra-high molecular weight polyethylene resin) should be low (0.1 wt.%) to avoid defects and achieve high joint strengths [97,98] while using Nd: Yag laser (λ: 532 nm in a single pulse, maximum pulse energy: 150 mJ). Studies by Rodríguez-Vidal et al. [45] showed that for low CNT concentration (0.01–0.05%) in the ABS, the process window increases significantly.

In the LTW of optical transparent thermoplastics, the cross-section of the HAZ becomes elliptical as the laser power is increased [188]. However, the elliptical or lenticular shape is asymmetric to the joining plane [189]. However, it is found that the weld efficiency increases for elliptical beam spots, and there is an increase in the weld strength for larger beam spot diameters [190].

These studies help us to understand that an optimum value of laser power enhances the weld strength, which may be due to an increased depth of penetration and weld width during LTW. The value for laser power is determined by considering material properties (polymer matrix and CB properties), beam diameter and scanning speed. Changing the value of laser power leads to a change in the shape and symmetry of the weld zone, but this needs further investigation. A concept of line energy helps to plot a characteristic curve for process behaviour visualization and determining process window location with certain limitations.

### 6.2. Scanning Speed

Faster production with better product quality is an essential aim of any manufacturing industry. For decreasing the cycle time, scanning/welding speed need to be increased. Sufficient time has to be given for melting and inter-diffusion of the two polymer surfaces. The temperature distribution is also crucial to achieving a high-quality joint [191]. Brodhun et al. [191] investigated the influence of different scanning parameters and scanning strategies for making a favourable temperature field using fibre laser (maximum output power: 70 W, spot diameter: 100 μm, λ: 1059–1065 nm). They utilised a fibre laser of λ 1059–1065 nm having a maximum power of 70 W along with an optical scanner. Laser joints were produced with one slow scan speed of 50 mm/s with one repetition and one fast scan speed of 3200 mm/s with 64 repeats with laser power of 18 W. A laser spot of 100 µm and 70 W was used to scan a circular area (d = 8 mm). Two strategies have been used to investigate the scanning path shown in Figure 12. Strategy A involved bidirectional scan path and strategy B involved a spiral scan. It is found that for strategy A, the maximum temperature moves from left to right with the material at the right becoming the most heated due to conduction. In the case of strategy B, the highest temperature is located in the spiral centre. The laser spot passes the centre of the scanned area several times in a short period of time due to a small radius. The central area gets damaged over a period of time. From the literature mentioned in the previous section, dispersion and distribution of CB in the absorbing component contributes an important role in weld integrity. It would be interesting to carry out studies using hybrid CB with varying ratios and studying the effect of this material system with changing scanning patterns. This type of research has not been carried out so far, but it would be interesting to carry out such analysis which can be generalised for understanding the LTW process.

To investigate the scanning speed influence and the number of repetitions on temperature distribution, 64-times higher scan speed (3200 mm/s) have been utilised (scanning the area 64 times). The faster scanning process and an increased number of repetitions lead to a homogeneous temperature distribution for the middle area. However, they found that the edges are not able to achieve the desired temperatures. In such cases, the design of the welded product has to be taken into consideration. This technique cannot be applied for sharp turn edges due to a lack of sufficient time for the material to melt at high speeds. Wang et al. [22] carried out laser trials using a compact 130/140 laser and a scanning galvanometer system (maximum laser power: 130 W, λ: 980 10 nm). They studied the effect of laser power and scanning speed on the joint width and related it to the weld strength. An increased weld width is observed at slow scanning speed and high laser power. At lower laser power, the number of scans has no significant effect on the joint width. However, with an increase in the laser power and scanning number, an increase in the joint width is observed.

This was further explained by Prabhakaran et al. [39] by analysing the effect of laser speed on polymer meltdown at a given laser power. They used 150 W CW diode laser having a focal distance of 100 nm and operating at 940 nm. As the polymer was exposed to heat for a short period due to increasing laser speed, the less molten polymer was generated. They found that with the optimised condition of power and speed weld strengths of around 70 MPa can be achieved concerning weld strength. Increasing the scanning speed causes a reduction in the maximum temperature [192]. The scanning speed significantly impacts the thermal defect zone size and joint strength [193]. Scanning speed and lap shear strength are inversely related to each other [24], irradiation time decreases as the scanning speed was too high. Lower scan speeds can enhance bond strengths due to an increased melt volume of the base material and weld-seam width. Amanat et al. [10] welded PEEK films using a pulsed fibre laser of λ 1060 nm and maximum power: 20 W. They used two laser powers: 10 W and 20 W and five scanning speeds: 4, 8, 16, 32 and 64 mm/s, to assess the joint quality for SC and amorphous PEEK. According to their results, the two lowest scan speeds, namely, 4 mm/s and 8 mm/s, showed the most significant bond strengths of around 22 to 25 MPa for SC and 12 to 19 MPa for amorphous PEEK. Increasing scanning speed may lead to a higher production rate. At the same time, lower values of scanning speed allow the development of good weld strength. However, material degradation is prevented at low scanning speed. During the studies on quasi-simultaneous laser welding of plastics, Routsalainen et al. [194] found that the weld strength (N/mm^2^) was nearly the same irrespective of the welding time or the number of scans used. For diode lasers operating at 940 ± 10 nm, the weld strength range was 31–32 N/mm^2^. They varied the scanning speed, number of scans, and laser power and also performed the experiments using the welding time as 0.5, 1 and 2 s by adjusting the values of scanning speed in order to fit the productivity requirements of mass production. Equation (7) [194] defines the welding time $t_{weld}$ (s) mathematically:(7)$$t_{weld}=\frac{n_{scan}\cdot l_{scan}}{v_{scan}}$$
where $n_{scan}$ is the number of scans, $l_{scan}$ is the scanning path length (m) and $v_{scan}$ is the laser scanning speed (m/s).

They used natural PC and commercial black PC as the laser transmissive and laser absorbing material respectively for overlap joint configuration. Figure 13 shows the effect of welding time to the weld strength with varying number of scans in the case of diode laser (940 ± 10 nm; Ø1.0 mm focal spot) welded samples.

Various optimisation and simulation techniques can be used to set the desired value of the scanning speed for decreasing the scrap rate and the number of laser iterations. The authors contend that to have sufficient polymer to melt at higher speeds, CB of an optimum thermal conductivity must be considered. This will allow the compounded joining parts to reach their melting point in a shorter period and provide sufficient polymer to melt to form sufficient weld strength.

### 6.3. Clamping Pressure

Sufficient clamping pressure is required throughout the welding process, and hence this pressure plays a vital role in LTW [24]. It is applied during the laser welding process to keep two surfaces in contact with each other and ensure the inter-diffusion phenomenon of the two surfaces. Wu et al. [169] explained the inter-diffusion of polymer surfaces, which leads to mechanical strength development, as mentioned in the previous section. Sometimes, voids as defects are formed during the laser welding process. These voids may be created due to an early release of the clamping pressure, i.e., before the inter-diffusion phenomenon followed by an insufficient cooling time of the melt. Lower clamping pressure may also result in elastic or plastic deformations of the product. The weld strength increases due to increasing clamping pressure [39]. A very high clamping force can also lead to distortion of the sample. Some compliance (elastomeric element) [11] is often used in the clamping system to present such distortion. A polymer containing a high dosage of CB in the absorbing part will melt at an early stage from a material perspective. In such cases, the clamping pressure can be set at a lower value to avoid distortion of the final product. The void formation can be due to the improper dispersion and distribution of the compounding ingredients such as CB. So far, there is no direct correlation that exists in the literature between clamping pressure and the material aspects. Hence, it would be interesting to systematically study various clamping force values with varying CB dosages and characterising the product.

Various clamping devices are used to hold the two surfaces during laser welding. Some of the clamping devices are placed within the laser beam path, which also serves as a guiding system along with clamping (Figure 14 (top)). In such cases, besides clamping, they also serve as a part of the beam guiding system. There are chances of clamping devices to get damaged due to their continuous interaction with the laser beam. Proper maintenance of these devices is necessary to prevent damage as it may affect the weld quality [189]. To avoid such situations in mass production, clamping devices are placed outside the beam path. To prevent the downtime risk and poor weld seam quality in the case of diode laser operating at 350 W and λ 940 nm, Devrient et al. [189] used a dual clamping device (DCD) (Figure 14) in which the clamping was carried out by a latch and an additional die (both made up of metal) which are placed outside the beam path. As seen in Figure 14 (bottom), the additional die prevents the deformation of the clamped parts in addition to reducing the stress. Such innovative design and techniques can help overcome some of the laser welding process problems.

For a good weld, parameters, namely, line energy, scanning speed, clamping pressure, etc., all need to be taken into consideration simultaneously. Multiple factors and their interactions have to be kept in mind to develop a good joint strength product. Huang et al. [195] studied joint strength based on line energy and clamp pressure interaction with the help of a response surface plot while using a 130/140 diode laser CW type for joining PC and PA66GF (glass reinforced polyamide). The response surface plot showed an increase in the joint strength with line energy at 0.11~5.00 J/mm and clamp pressure at 0~0.4 MPa. When the parameters were increased beyond these values, line energy at 5.00~9.00 J/mm, and clamp pressure at 0.4~0.8 MPa, there was a decrease in the joint strength. Energy density increased with an increase of the line energy at constant spot diameter and a given clamp pressure. Apart from uniform dispersion and distribution of CB, an increased joint strength may also be due to the higher Vander Waals forces. As discussed, different types of clamping unit designs and proper process parameters can prevent the downtime risk or poor weld seam quality resulting from contaminated clamping devices or improper clamped joining partners

## 7. Performance Characterisation and Weld Quality

Polymer characterisation before and after laser welding provides an insight into optimising and improving our understanding of material properties such as thermal stability, mechanical strength, rheological, optical, etc. and its performance. Characterisation of commercial materials can be extremely complex as it requires skills for component analysis as well as in their deformulation. Laser welded samples are difficult to characterise if they have complex shapes. Besides, selection and extraction of the laser-welded polymer material and HAZ are quite challenging. This section provides guidelines for weld quality assessment through various standards. It emphasises the structure-property relationship of SC polymers after melting and recrystallisation [196].

### 7.1. Flow Behaviour

MFI determination is specifically relevant in the design of processing equipment, mould design and optimisation of process parameters [197]. MFI determines the ease of flow of the polymer melt. Kuriakose et al. [197] carried out melt flow measurements using a capillary rheometer while studying the melt flow behaviour of thermoplastic elastomers from PP–Natural Rubber (NR) blends. PP copolymers, having a small percentage of ethylene content, have less viscosity [198]. The addition of fillers increases PP viscosity, while plasticiser and compatibiliser decrease the viscosity. A higher MFI value results in a higher degree of crystallinity. MFI plays an essential role in achieving high weld strength [38]. An increase in polymer crystallinity leads to an increase in the MFI value. PP has a higher crystallinity of up to 90% as compared to HDPE (high-density polyethylene), 85%, and LDPE (low density polyethylene), 60%. Due to this reason, it is more difficult to have higher values of weld resistance for LDPE as compared to HDPE. Table 3 provides a reference value for the MFI of the PP samples according to their molecular weight. The relation of MFI and crystallinity of LT and LA part should be taken into consideration while selecting the laser parameters for better inter-diffusion for achieving high weld strength

### 7.2. Thermal Studies

Differential scanning calorimetry (DSC) is a thermo-analytical technique that enables the determination of various transition temperatures: T_g_, T_m_, T_c_, and mesomorphic transition temperature [199]. DSC also determines the corresponding changes in entropy and enthalpy. It is essential to determine these parameters before material processing and to decide the laser welding parameters. It also helps to predict the weldability of various polymers. Juhl et al. [60] tried to predict the laser weldability of dissimilar polymers using Laserline diode laser of a maximum power of 30 W and λ = 808 nm for high-density polyethylene (HDPE), PP, poly(methyl methacrylate) (PMMA), polystyrene (PS), poly(butylene terephthalate) (PBT) and polycarbonatre (PC). LT part was welded to the laser absorbing part containing 0.4% CB. They used Q2000 DSC from TA instruments to determine T_g_, T_m_, and T_c_ [60]. They also determined the difference in T_m_ and Tc values when adding CB to the pre-resin. They observed the greatest strength of the polymer weld when the absorbing polymer has the highest melting point of the two welded polymers (relative weld strength of transmissive PC and absorptive PBT, T_m_ 224.7 C, the combination was 1.06 while the combination of PC welding together was only 1). They also reported weldability of these combinations: HDPE/PP, PMMA/PC and PS/PC, which were regarded as “not weldable” in the industrial chart (Table 4).

Thermal transitions are often studied to identify and better understand the welding process and the correlation with the weldability of the material. Thermogravimetric analysis (TGA) is a technique to determine the temperature at which the material degradation occurs. The significance of TGA/derivative thermogravimetry (DTG) is to measure the thermal stability of the material undergoing laser welding. In addition, one can analyse the effect of changing the type of CB or its loading on thermal stability and set the laser parameters accordingly. DTG (derivative plot of TGA) helps to understand the decomposition of the constituents presents in an organic matter at a given temperature through its peaks. Char percentage would be given in the TGA analyses, which is higher for aromatic polymers. Polymers with aromatic content are often not suitable for laser welding as the aromatic chains oppose the formation of entanglements due to an increased chain stiffness [200,201]. The temperature at the weld interface influences the weld strength in LTW. The weld strength increases with an increase in temperature to some extent. However, it may also lead to the accumulation of material degradation in terms of weld defects (voids), leading to a decrease in weld strength. [202].

Pelsmaeker et al. [123] characterised various polymers (PA6, HDPE, PLLA, PP, etc.), optically and chemically, to relate the intrinsic properties (intrinsic viscosity, chain stiffness, etc.) with the welding performances [123]. The experiments were performed on a Thulium fibre laser of λ 1940 nm of average power 120 W; the concept can be utilised for other diode lasers. They utilised Q2000 DSC for determining parameters such as T_m_, melting enthalpy (ΔH_m_), the heat of fusion (ΔH_fus_), degree of crystallinity, etc., to estimate the probability of successful welding. Some conclusions were also drawn to the crystalline domains. Although PA66 is transparent in appearance, it exhibited a degree of crystallinity of 30.4%. Crystalline domains may also oppose the welding efficiency of two polymer layers like chain rigidity observed in aromatic polymers. There was unsatisfactory welding observed for these two polymers with 49.6% of crystallinity for PP and 86.7% for HDPE. PLLA with a degree of crystallinity of 9.4% resulted in stronger welds. The reduced welding performance could be due to the scattering of the incident laser light on the crystalline domains. According to this study, amorphous polymer chains gain sufficient mobility upon laser beam and result in strong welds. Thus, it becomes crucial to control the crystallinity of the polymer by carefully selecting the compounding ingredients and their processing conditions like extrusion, injection moulding, etc. The degree of crystallinity which leads to poor welding differs from polymer to polymer and needs further investigation.

In the case of laser-welded products, one of the areas of the mating surfaces undergoes an additional heating history compared to the adjacent polymer, which does not experience this heating. During the LTW process of SC polymers, the material in the HAZ region gets heated up to the T_m_ and then cools down (recrystallization). During this process, different polymer morphology can be formed, which is related to the weld properties. These changes can be studied through DSC by measuring the percentage crystallinity of the polymer before and after laser welding at various laser parameters [203]. In the review article on measuring the polymer crystallinity using DSC, Kong et al. [203] mentioned that the degree of crystallinity changes due to crystallisation, partial melting and annealing re-crystallisation, and complete melting takes place while heating the sample to the melting point. It is advised to determine the specific heat temperature dependence separately from the amorphous samples or the literature. This technique has a broad dynamic range concerning its heating and cooling rates that includes the operation of isothermal and temperature modulated. According to Schawe [204], there can be errors and uncertainties in the enthalpy determination if a complicated measuring program and evaluation procedure of temperature modulated DSC is used [204,205].

However, the DSC graph provides useful information such as a shift in the T_g_, indicating a change in mobility of the polymer chains or a change in the neighbouring molecules/chains. There are specific challenges in DSC graph interpretation that require expertise. Sometimes there can be multiple peaks indicating two crystal structures of the same material or two different materials. Chen [203] mentioned that the real degradation point would be generally expected to be higher than the measured value by TGA due to a very short heating period in the case of high-speed contour welding. In his studies, the scanning speed of 25 mm/s was used for about 0.02 s using the laser head Rofin-Sinar DLx16 CW 160 W diode laser (λ 940 nm) and workstation UW200.

Higher laser power or scanning duration is required for an increased polymer melt facilitating stronger weld strength. At the same time, the polymer matrix should not be degraded due to high temperature. With an increase in the amount of CB, the thermal stability of the material increases. Verma et al. [206] prepared nanoscale composites (NCs) of PPCP with multiwall CNT(MWCNTs) and examined the thermal stability of NCs by using TGA/DTG. About 8 ± 1 mg of sample was taken in a Perkin Elmer Pyris 6 TGA with a heating rate of 20 °C/min in N_2_ atmosphere (flow rate 60 mL/min). With an increase in the amount of MWCNTs, there is an increase in initial degradation temperature values, the temperature at the maximum mass loss rate, and the final degradation temperature. The overall thermal stability increased with the carbon content. It is believed that CNTs facilitates heat dissipation within the nanoscale composites and prevents any local heat accumulation, which may increase initial degradation temperature, final degradation temperature and maximum mass loss rate [123]. During the electromagnetic interference shielding studies, Verma et al. [206] found no loss for MWCNTs for temperatures up to 500 °C for PPCP, but at 500 °C, the mass losses were more than 99.82%. It is important to understand the degradation of CB added to the polymer matrix for deciding the LTW operating parameters. The operating temperature should be lower than the degradation temperature.

There are different environmental conditions in which TGA analyses can be carried out. The TGA chamber may be under air, oxygen, argon, nitrogen, etc., which depends upon the purpose of the research or study. An inert atmosphere is chosen to avoid oxidation/reduction/side reactions. Pure oxygen at high temperatures may also damage the instrument. Nitrogen and argon do not interfere during degradation but can form nitrides for inorganic compounds. Hence, various possibilities are to be considered while carrying out TGA analysis, depending upon the material used. For laser-welded analysis, sample preparation from the HAZ area, using a microtome tool, is quite challenging.

### 7.3. Morphological, Microstructure and Phase Studies

The morphology of the laser-welded parts ultimately influences the weld integrity of the product. Various microscopic techniques can be utilised to study the crystal type, spherulitic structure, aggregates, agglomeration of CB, uniformity of its dispersion, etc.

The HAZ of various laser welded products can be identified by examining the weld cross-section irradiated by a polarized light using a microscope. Chen [203] investigated the HAZ (prepared using microtome) for PA6 containing various dosages of CB (0.1, 0.025, and 0.0668 wt.%). Figure 15 shows the HAZ observed through the polarized light microscope for 0.1 wt.% CB along with the contact model.

There are various optical microscopes, such as Olympus STM 6, which can provide high-performance three-axis measurements of parts along with sub-micron precision [207] which can be used to take measurements such as weld seam widths [36].

Microscopic techniques can also be used for analysing various welding defects and material flow. Liu et al. [208] used a novel method of welding by incorporating metal as an absorber to obtain a clear and high-quality joint using a diode laser (maximum power: 50 W, λ: 808 nm). The internal micromorphology of joints welded with Fe wires (or CB) under given input power and clamping pressure was observed. Figure 16 shows the optical micrographs of joints in Fe wire (upper) and CB groups (lower) with different input powers (feed speed, V = 1 mm/s and clamping pressure, C = 0.4 MPa). In the case of the Fe wire group for no dye absorber, a series of clean and impurity-free joints are seen whereas there is the presence of contamination in CB joints with CB powders. The highest temperature occurs on the surface of Fe wires due to higher optical absorption of Fe as compared to PET. Bubble formation (gas monomers) occurred for an input power of 2.7 W and 5.5 W for CB groups and matrix containing Fe wire, respectively. The bubbles are present in the middle of the welding seam where the polymer has decomposed in the case of the CB absorber. At 4.4 W, these grew to be a vent-hole for more intension of polymer thermolysis. Beyond 4.4 W, there is burning and smoking of the material (for Fe wire containing matrix, this happens beyond 7.1 W).

The joint strength of the weld is dependent upon the molten depth to a certain extent. Due to the presence of CB in the absorbing part of the welding system, the molten depth value is expected to be more as compared to the LT part. However, the results obtained by Wang et al. [183] were different. They investigated the relationships of process parameters, molten pool geometry, and shear strength in the LTW process of LT layer as polyethylene terephthalate (PET) and LA layer as PP (maximum laser power: 130 W and λ: 980 ± 10 nm) [183]. They measured the molten depth of PET and PP considering their Vicat softening point, 85 °C and 140 °C, respectively, and weld width under a three-dimensional VHX-1000 microscopy. Some of the simulation studies show that the penetration depth in the transparent polymer is always less than that in the absorbing polymer [209,210]. However, through the microscopic images (Figure 17), they found that the molten depth is greater in PET than in PP. The same was confirmed with simulation results. This is due to the thermal conductivity of the PET. The semi-oval profile of the weld zone was also observed through the microscope (Figure 17). For analysis, the samples were scrubbed with acetone to remove dirt or stains from the polymer surface, followed by ultrasonic cleaning and finally drying for 12 h for dust and water removal. Three-dimensional VHX-1000 microscopy made by KEYENCE was used for measuring the molten pool geometry (weld width and molten depths).

Apart from molten depth, weld seam width is also studied carefully by various researchers to define joint efficiency. For such minute examination, a microtome is often used to slice the welded areas in the micron range thickness [34]. Kiss et al. [211] carried out a microscopic analysis of the morphology of seams in friction stir welded PP using optical and electron microscopy. They used Olympus BX 51 (Tokyo, Japan) optical microscope in transmission mode and different types of microtome (10 μm thick samples, Bright 5040 microtome and 3–5 μm Leica EM UC6 ultramicrotome) for analysing the samples of thickness 10 µm and 3–5 µm sliced from the centre of the seam, base material and borderline between the two zones. They compared one sample with maximum tensile strength welded at a rotation speed of 3000 rpm and another with lower welding tensile strength which was welded at 2000 rpm. They found that the joint efficiency (strength of the welded joint concerning the strength of the base material) was better for smaller seam width as compared to larger seam width with complex morphology (distorted spherulites formation under shear stresses in the HAZ where PP became softer). The microscopic analysis helps to analyse the various types of defects. Similar morphological studies were carried out by Vidal et al. [45] while carrying out the LTW of transparent ABS and ABS filled by varying percentage of carbon nanotubes (CNTs) using a diode laser system (maximum output power: 50 W, λ: 808 ± 4 nm). They presented micrographs of the weld surface through scanning electron microscopy (SEM) to analyse material decomposition, degradation, melting zone and weld porosity of a polished cross-section of the welded product. The weld widths and integrity of the top and joining cross-sections of the weld area can also be studied through microscopic characterisation. Micromorphological studies were carried out to analyse the post-failure joints for bubble inspection after the tensile tests. Figure 18 shows SEM images of welding seams at various sections of Fe wire and CB group after undergoing tensile testing with V = 1 mm/s and C = 0.4 MPa with an input power of 3.8 W [208]. A1 and A2 show a good connection or adherence between PET and Fe wire due to the presence of tiny bubbles. According to Liu et al. [208] (using a diode laser of maximum power: 50 W, λ: 808 nm) and Miyashita et al. [212] (using fibre laser of λ 1070 nm operated at 280 W), the bond strength of the samples having numerous and tiny bubbles are higher compared to those having large or no bubbles [212].

Ghorbel et al. [49] investigated the influence of the process parameters (laser power 20–40 W and scanning speed 3–6 mm/s) on the weld zone geometry and the weld zone [49]. The thin slices of the samples were etched in a H3PO4/H2SO4 solution for 20 h at 20 °C to examine the HAZ under the SEM [213,214,215,216]. They observed well-distinguished zones in the welded part. The thermal cycle or the thermal history of the material determines the crystalline morphology, as already discussed in Section 3.2 (Figure 4). Further studies are still needed to get a more in-depth insight into the modification of the polymer microstructure and its evolution with the process parameters. SC polymers are complex systems, microstructurally and mechanically. Correlating the spherulite dimension change with the mechanical behaviour of the weld seam is quite challenging.

Another theory for CB distribution could be: for a relatively low concentration of CB, all of the available laser beam energy will be absorbed close to the interface. With an increase in the CB content, the depth of the initial energy deposition (penetration depth of the laser beam) may be smaller, resulting in the heating of smaller volumes with higher initial temperatures. The depth of penetration remains smaller than the final weld depth irrespective of the CB concentration. The LT material, along with its surrounding material, gets heated by conduction. The rate of conduction depends on the temperature difference between the initially heated volume and the surrounding volume. For higher CB concentration, the initially heated volume temperature will be larger (all of the laser energy, but in a smaller volume due to smaller penetration depth), which will drive heat more quickly into the surrounding areas. In general, with an increased CB, there will be increased thermal conductivity so that heat will be dissipated more quickly away from the initially heated volume. It can be envisaged that, in some cases, this could result in wider, deeper weld zones and in other cases where the thermal rate of conduction is so high as to dissipate the laser energy too widely to the surroundings, weld zones could be narrower and shallower.

The dispersion and distribution of CB in the polymer matrix can be examined through transmission electron microscopy (TEM). TEM facilitates a beam of electrons to transmit through the sample and form an image. It helps to study morphology (shape, size, and distribution of particles on the scale of the atomic diameter), crystallographic information (arrangement and order of atoms and their defects), and compositional information. Grimberg et al. [217] used TEM to characterise the structure (microstructure, grains, etc.), the interface morphology, and the composition of TiN coatings on metallised ABS [217]. Verma et al. [62] also utilised TEM for determining the nanoscale dispersion of CNTs. For TEM analysis, samples were prepared using a Leica Ultramicrotome (30–80 nm thick). Through TEM images (Figure 19), they were able to visualize the formation of the interconnected or network-like structure of MWCNT, which restricted the polymer chains mobility. The morphological characterisation of the melt blended samples using TEM showed that there was a uniform dispersion of CNTs in the matrix of PPCP. TEM is a preferred method for measuring the CB morphology [218]. ASTM D3849-14a provides a standard test method for CB morphological characterisation using electron microscopy. The analyses of dispersion and distribution of the nucleating agents in the LT layer and fillers in the LA layer can form a basis to understand the welding process, which remains an unexplored field using TEM.

In the case of SC polymers, the morphology of crystals and crystallinity can be directly correlated with engineering and physical properties [219]. Some studies showed that the different chemical structures of nylon-6 (α- and γ-forms) provide different mechanical and physical properties. The two forms differ by 53 and 50% in hardness and elastic modulus, respectively. The difference in the packing density of the polymer chains and the strength and number of the hydrogen bonds may have led to these differences. It was stated that with an increase in the percentage of crystallinity, there was an increase in the modulus, hardness and density [220]. In these polymers, the incident laser beam interacts with the crystal structure and propagates according to the polymer morphology [221]. As previously discussed, the morphology depends upon the type of nucleating agents or the compounding ingredients used in the formulation and the processing conditions. Hence, for LT parts, the scattering coefficient plays an important role [222]. It is not desirable to have higher scattering for a good weld seam quality. Laser welding of the SC material is challenging because, with an increase in crystallinity, the material can change from transparent to semi-transparent and opaque [223].

X-ray diffraction (XRD) is a powerful and rapid technique for identifying unknown crystalline materials. The data interpretation is often quite straightforward. The degree of crystallinity can be estimated by comparing the diffraction lines of the given sample with the broad halo of the amorphous regions. Crystallite size can be calculated using the Scherrer Equation (8) [224,225,226,227,228]:(8)$$\Phi =\;\frac{K\lambda }{\beta cos\theta }$$
where

*Φ:* crystallite size (m)*K:* shape factor (≈1)*λ:* wavelength of x-ray (m)*β:* full-width half-maximum (FWHM in radians) of the most intense peaks*θ:* Bragg angle in radians

The shape of the crystal and the size distribution determine K. It varies from 0.62 to 2.08 [229]. One of the studies of PLA could form a basis to understand the effect of crystallinity on the mechanical properties of SC polymers. Perego et al. [230] studied the effect of crystallinity and molecular weight on poly(lactic acid) mechanical properties. The mechanical properties of PLA (with reduced D-lactide isomer content) were improved by the fibre stretching technique. Mueller et al. [231] found that the film fibre with more crystallinity has higher tensile strength. However, CB used in the LA layer also has a nucleating effect, as observed by Mucha et al. [86]. Such polymer morphology modification can result in a total collapse of the polymer spherulite structure [232]. Hence, the crystallinity of the polymer sample should be measured after the final processing of the fabrication process.

### 7.4. Mechanical Properties

The SC polymer consists of amorphous as well as crystalline phases. The microstructures in the crystalline phase have a close correlation with mechanical properties. The spherulites with larger sizes exhibit crack at their boundaries, whereas those with smaller sizes are smooth [125]. These observations are generally for polymers crystallised from the melt as thin films. It is believed that the boundaries of the spherulites are effective sites for fracture due to the presence of the foreign matter layer [233,234]. Many studies were carried out to analyse the effect of spherulite diameter with yield stress. Reinshagen and Dunlap [235], reported that the value of yield stress decreases with an increase in spherulite diameter. However, Kleiner et al. [236] found that with spherulite diameter, yield stress increases. The authors contend that the smaller size of spherulites in the weld zone would result in better mechanical stress distribution. However, to clarify the mechanism involved, further investigation needs to be conducted. There are various levels (three important levels) of microstructures associated with the crystalline matrix that controls some of the polymer properties [125]. The amorphous phase behaviour is controlled within the 1–10 Ǻ level [237]. Defects present in the crystalline phase increases the inter-chain spacings. Sliding of the chains becomes easier as the spacings are increased, which ultimately lowers the crystals shear strength [238]. The second significant level of the microstructure is of the order of 100 Ǻ where the crystals are stacked one above the other with amorphous phase in-between. There are tie-chains that lies in the amorphous region and joins the two crystal lamellae. These tie chains control the elastic modulus level [239] and the yield strength of the polymer [240]. Voids formed during the polymer crystallisation can also act as mechanical stress raisers as well as water-saturated sites leading to an electro-mechanical breakdown.

The failure mechanism and path chose (crack propagation) by the polymers depend mainly upon chain chemistry, molecular weight and degree of imperfection, loading conditions, and thermomechanical history. In reality, the properties are often correlated with microstructural detail and phase behaviour. These are ultimately controlled by the chain, processing history, and the test condition variables. Laser welding parameters (laser power, scanning speed/welding speed, clamping pressure, spot diameter, etc.), polymer formulation, and processing determine the mechanical properties of the laser joint [24,241].

In general, increasing the CB content increases the mechanical properties of the polymer matrix [242]. This is true up to an optimum amount of CB, beyond which, the material becomes brittle in nature. Huang et al. [242] prepared polycarbonate PC-PP blends and studied the effect of CB (Vulcan XC 72R CB from Cabot Corporation) on their mechanical properties. It is a conductive CB grade for thermoplastic applications having a surface area of about 254 m^2^/g. The dosage required for conductive CB in the laser absorbing part is relatively less. There is an increase in the tensile strength, tensile modulus, and flexural modulus with CB dosage (amount varies with polymer matrix type, processing conditions, etc.). In contrast, elongation at break, impact strength and toughness decreased. Vidal et al. [45] carried out trials using a diode laser machine with maximum output power: 50 W and λ 808 ± 4 nm). They studied the effect of concentration of CNT (0.01% and 0.05%) on the mechanical properties of ABS during the LTW process. They used an Instron 3369 Static Universal testing machine for testing a dumbbell-shaped specimen complying with ASTM D638-82a Standard. The mechanical properties (cracking strain, tensile force, shear strength) increased by adding a certain amount of CNT (0.05 wt.% for the present studies). They also checked the mechanical resistance of the optimal welded joints by mechanical shear tests. Below a threshold value of the line energy, there was a direct relationship between the weld width and the shear force for both the CNTs concentration (0.01% and 0.05%) of ABS up to 0.27 J/mm.

Besides the material aspect, various types of weld joints provide different weld strength to the final product. The laser-welded structures experience dynamic service loads. Lap joints often experience defects such as notches, surface cracks, residual stress, etc. [243]. Howevert joints are often not preferred due to a smaller interfacial surface area. However, it is a simple joint and can be strengthened relatively easily.

Amanat et al. [10] defined various types of failure mode in case of lap-joint geometry of unfilled PEEK films of 250 μm (250a amorphous and 250c SC). These classifications are based on adhesive joints; however, they can be applied for welded joints. The modes of failure are based on adhesive-based bonding with an adhesive which is an additional material involved. For LTW, there is no adhesive, but there is the formation of HAZ which can be located outside the weld, which may lead to failure away from the joint. Figure 20 shows the optical micrographs of the joints before carrying out the mechanical tests with varying power and scanning speed. Heat damage of the welded part with power 20 W and speed 4 mm/s is also shown in Figure 21. In the case of crystalline samples, there was no observation of material damage. Figure 21b distinguishes between the bond width and the HAZ. They observed a prominent HAZ at slower speeds and no evidence of HAZ for the faster speeds.

Detailed analysis of the failed samples provides us with more information regarding the type of failure mechanism for LTW. There were mainly two types of failure: interfacial and substrate. If the joint fails immediately at the interface between the opposing surfaces, this term is defined as interfacial failure. Substrate failure occurs when the joint fails within the substrate and the bond are still intact (Figure 22). The substrate type of failure is again categorised into type I and type II failure (Table 5). In the type I failure, the substrate yields and failure occur away from the interface (most desired result), whereas, in type II failure, the bond remains intact. Still, the substrate fails proximal to the interfacial region.

Further studies are needed to develop mechanical performance data and optimise various processing conditions for non-black thermoplastic products, specifically, concerning the influence of pigments (on laser-welded samples) used within polymer compositions. Chain orientation, chain length, configuration, degree of polymerization, number average and weight average molecular weight can also enhance the mechanical properties of the laser-welded samples [244].

## 8. Conclusions and Future Trends

Polymer composites are extensively used in various engineering fields. Using a laser welding technique for fabrication of polymer composites will overcome the limitations concerning intricate product design requiring specific quality standard requirements.

In this review, a wide range of aspects related to laser weld technologies concerning weld quality has been presented. For the polymer formulation, various types of nucleating agents and CB concentrations are being used to control the transmissivity and absorptivity of the laser welding parts, respectively. CNTs, CFRP, BN and hybrid fillers are emerging fillers that can be used to achieve a strong weld joint. Welding of the non-black product with an optimised parameter is still unexplored. More studies are needed to evaluate the weld quality or specific energy changes due to additives like nucleating agents and combinations of fillers. These additives have an impact on the laser transmissivity or absorptivity of the laser beam. There are limitations in the experimental studies regarding the generalisation of the result. Also, the degree of crystallinity which correlates to weld strength differs from polymer to polymer and needs further investigation.

Optimised laser parameters like laser power, scanning speed and number of repetitions are to be selected with modelling and simulation software. Increasing the number of scanning repetitions favoured in maintaining homogeneous temperature distribution, but the desired temperature cannot be achieved in the edges of the product. Changing product design geometry or altering the surface chemistry of the polymer (plasma treatment or gamma radiations) may enhance the homogeneity of temperature distribution. Detailed understanding of surface chemistry provides an economical and specific approach for the rational design of laser weldable parts with desirable properties in the future. Various types of weld joint (lap joint, butt joint, etc.) and innovative clamping devices utilised in the laser welding process can improve the weld seam quality and increase the joint’s mechanical strength.

LTW is an expensive technique as compared to other joining methods considering the capital equipment investment. However, the overall cost per assembled sample (successful delivery and selling), makes LTW the most cost-effective industry technique. The surfaces to be welded should be clean and requires a good fit-up. For welding non-pigmented materials, some absorbent form will be needed for the laser beam at the joint interface, which adds an extra complication to the process [8]. There are various challenges and areas for optimisation such as fixture designing for welded product testing, obtaining high ingress protection ratings, for welded products etc.

Regardless of the significant work progress already carried out in the laser field, integrating the various available technologies is complex and challenging to achieve. The consumer businesses which have taken advantage of the connective technology can be scrutinised to get an idea of how the laser welding of composites could unlock the potential of Industry 4.0. Hence, the effective implementation of Industry 4.0 technologies is difficult and still a subject of research.

## Figures and Tables

**Figure 1 polymers-13-00675-f001:**
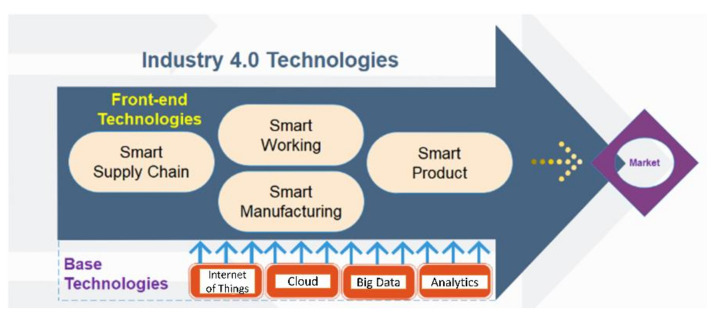
Theoretical framework of Industry 4.0 technologies [15]. Reproduced with permission from Ref. [15]. Copyright (2019), International Journal of Production Economics.

**Figure 2 polymers-13-00675-f002:**
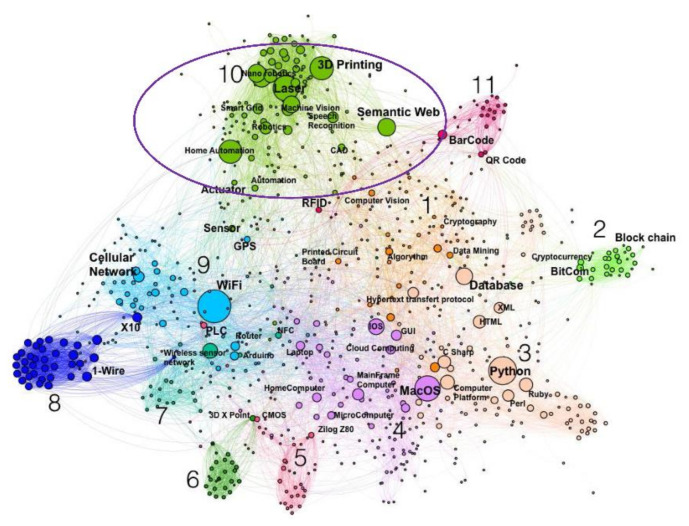
Graphical representation of Industry 4.0 technologies and their arrangement in the form of clusters [18]. Reproduced with permission from Ref. [18]. Copyright (2018), Computers in Industry.

**Figure 3 polymers-13-00675-f003:**
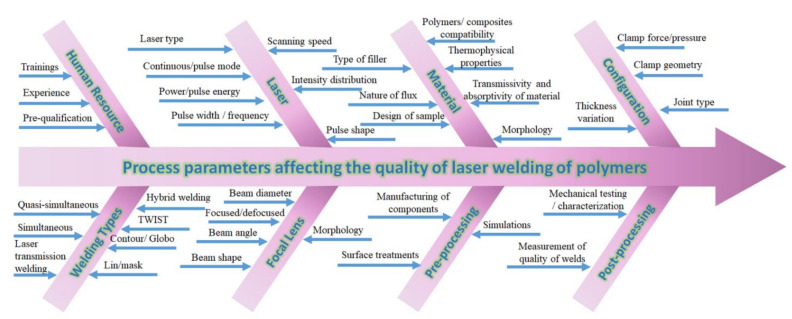
Ishikawa diagram representing the various factors influencing on quality of laser welding of polymers. Authors’ self-created diagram.

**Figure 4 polymers-13-00675-f004:**
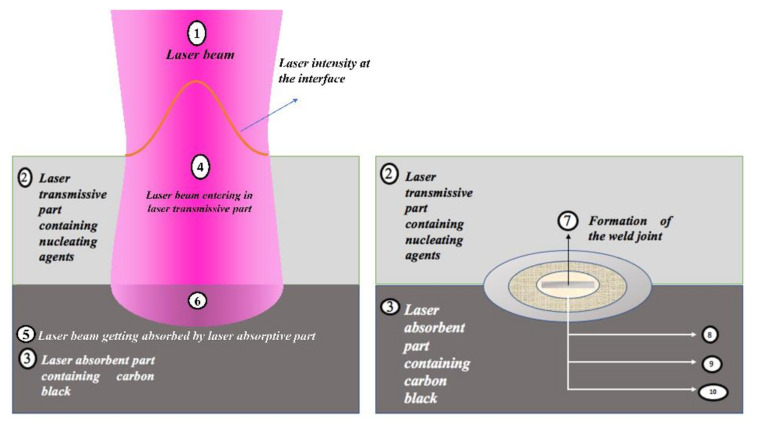
Lap joint and interaction of the laser beam during the LTW process.

**Figure 5 polymers-13-00675-f005:**
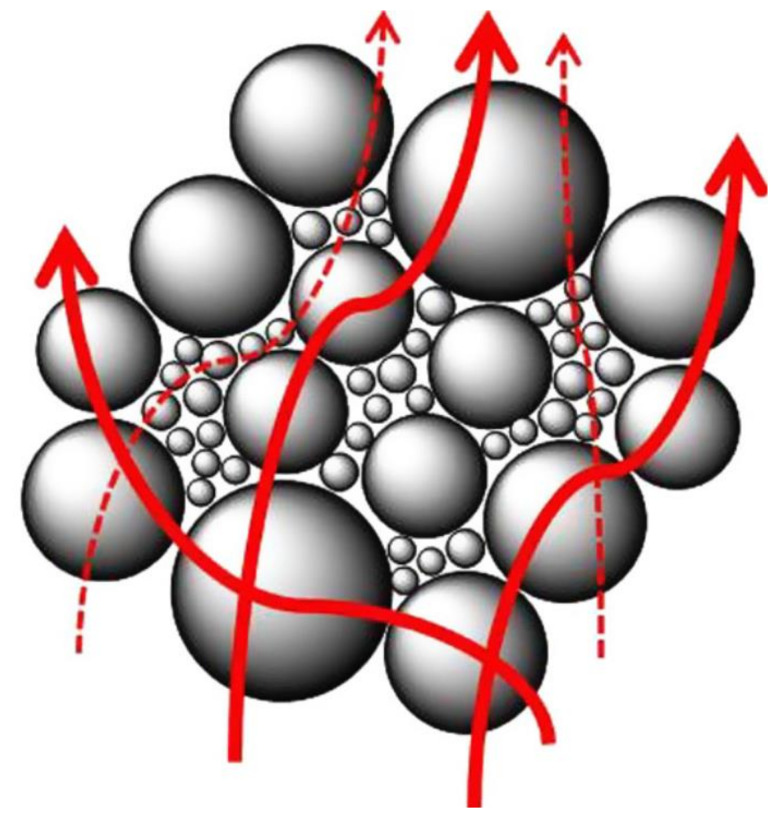
The concept of using hybrid filler particles resulting in more efficient fillers packing [96]. Reproduced with permission from Ref. [96]. Copyright (2013), Composites Part B: Engineering.

**Figure 6 polymers-13-00675-f006:**
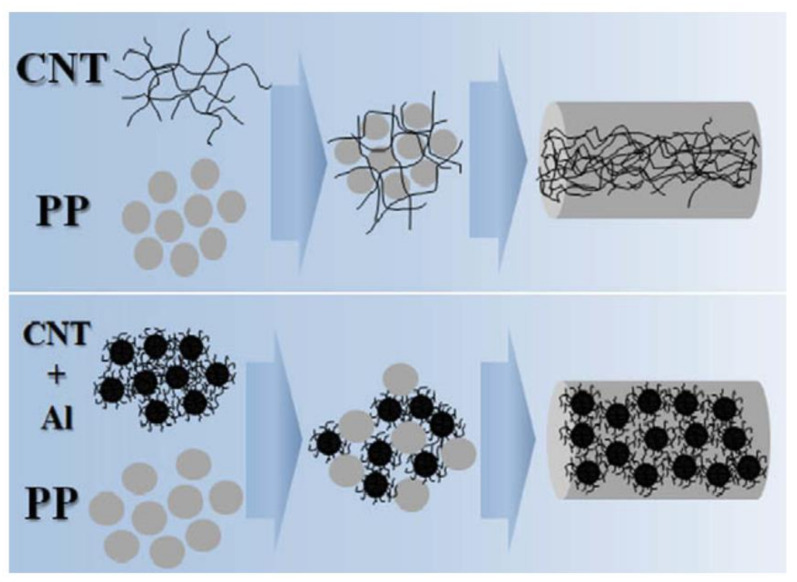
The schematic diagrams showing the dispersion of CNTs into the PP matrix [97]. Reproduced with permission from Ref. [97]. Copyright (2010), Composites Part A: Applied Science and Manufacturing.

**Figure 7 polymers-13-00675-f007:**
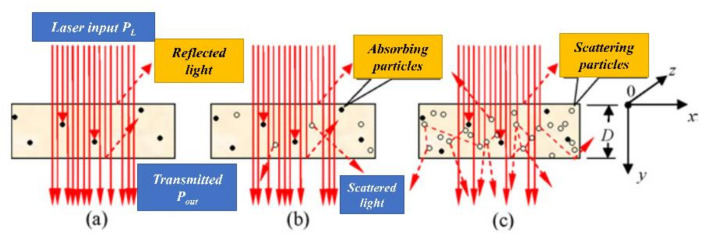
Schematic view of the laser beam transmission process in (**a**) light-absorbing non-scattering polymer; (**b**) light-absorbing single-scattering polymer; and (**c**) light-absorbing multi-scattering polymer [102]. Reproduced with permission from Ref. [102]. Copyright (2011), Journal of Materials Processing Technology.

**Figure 8 polymers-13-00675-f008:**
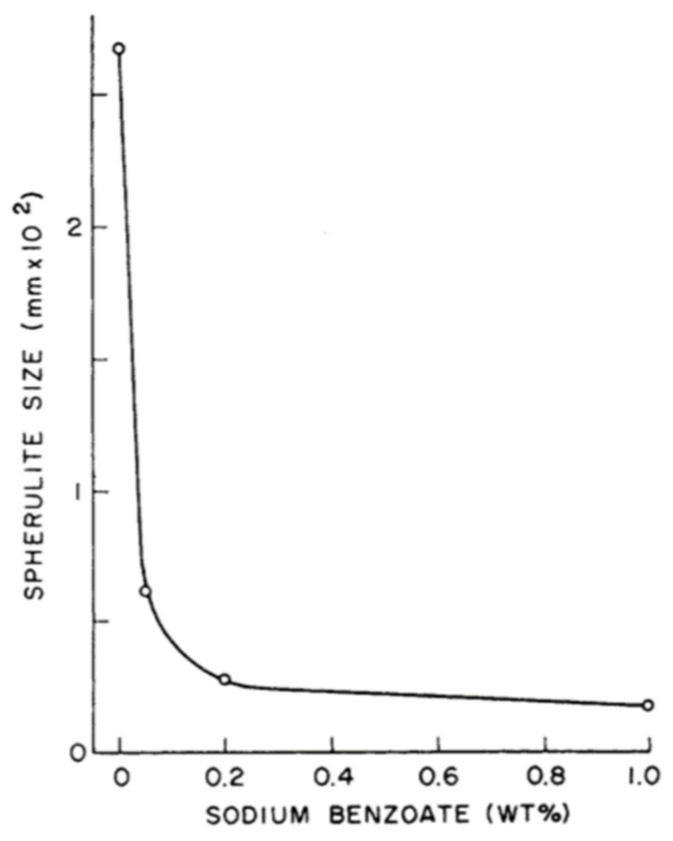
Spherulite diameter refinement of iPP through the addition of nucleating agent, sodium benzoate [117]. Reproduced with permission from Ref. [117]. Copyright (1984), Polymer Engineering and Science.

**Figure 9 polymers-13-00675-f009:**
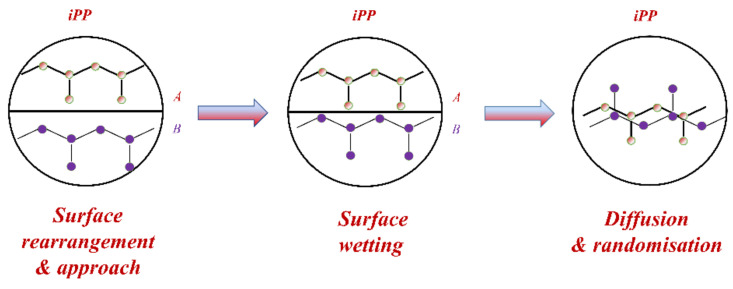
Mechanisms involved in self-healing via molecular inter-diffusion (Authors’ self-created diagram).

**Figure 10 polymers-13-00675-f010:**
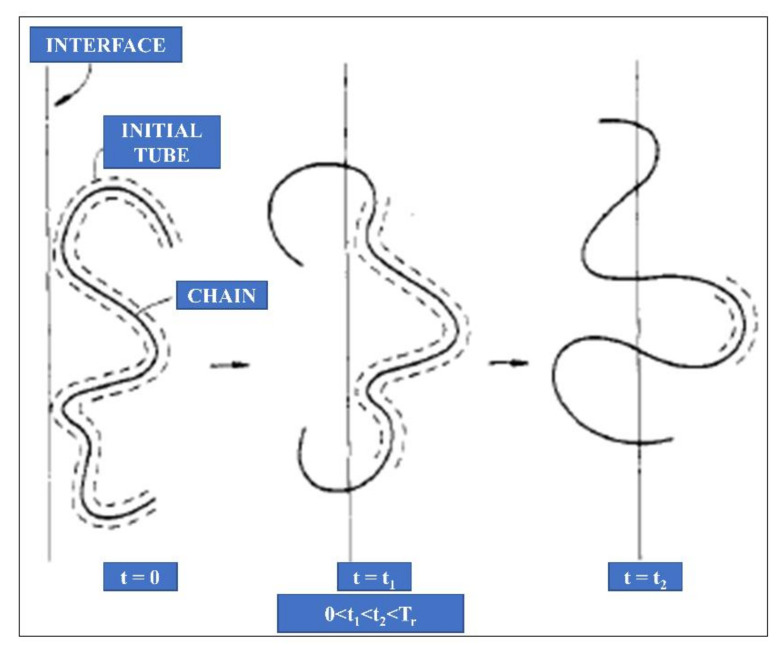
Polymer chain disentanglement from its initial tube near the interface [161]. Copyright (1981) American Chemical Society.

**Figure 11 polymers-13-00675-f011:**
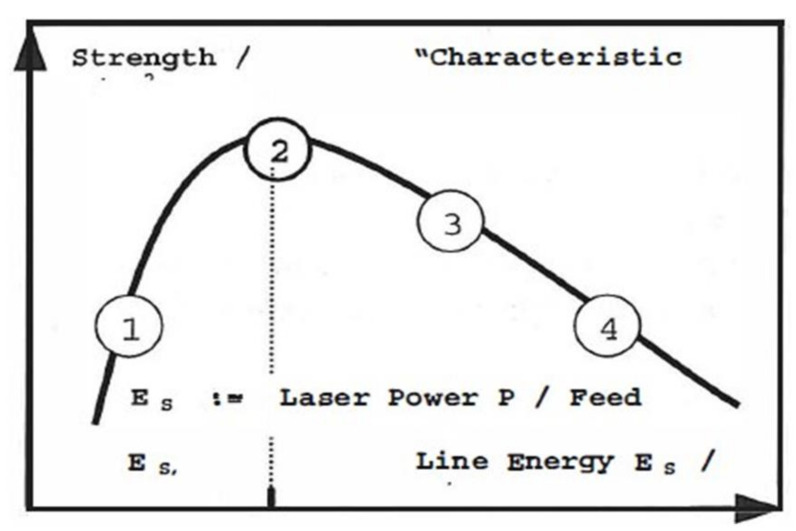
Characteristic curve—strength Vs line energy: 1. just light adhesion—no good welding; 2. optimal welding adjustment; 3. optimal adjustment left—bad welding; 4. decomposition—bad welding [176]. Reproduced from [176], with the permission of the Laser Institute of America.

**Figure 12 polymers-13-00675-f012:**
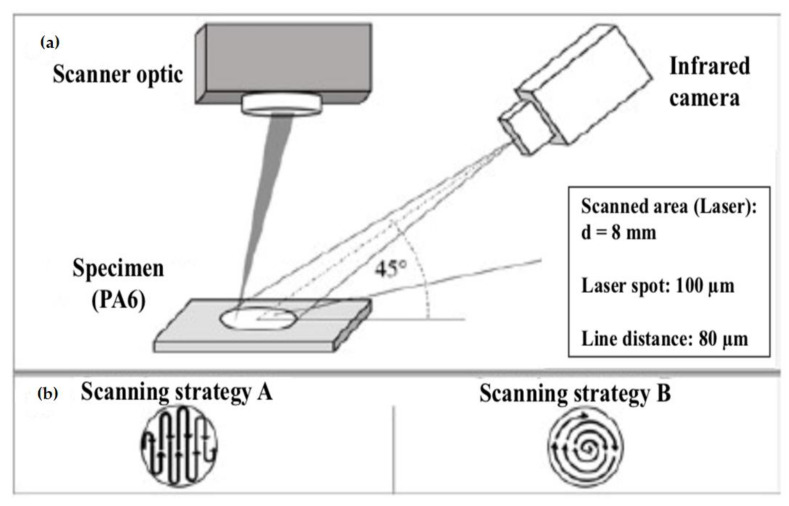
(**a**) Experimental set-up to analyse the temperature distribution; (**b**) Investigated scanning strategies, A (bidirectional) and B (spiral) [191]. Reproduced with permission from [191]. Copyright (2018), Procedia CIRP.

**Figure 13 polymers-13-00675-f013:**
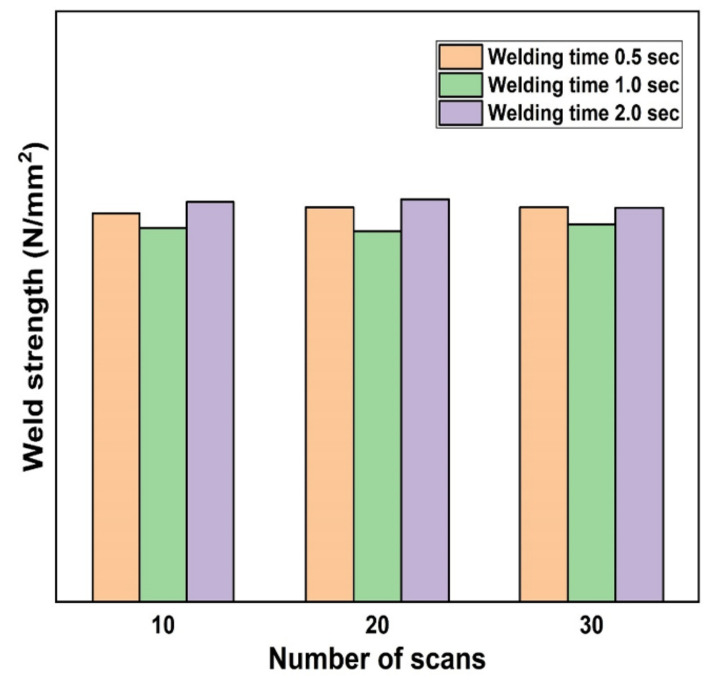
The effect of welding time on the weld strength with varying number of scans in case of diode laser [194].

**Figure 14 polymers-13-00675-f014:**
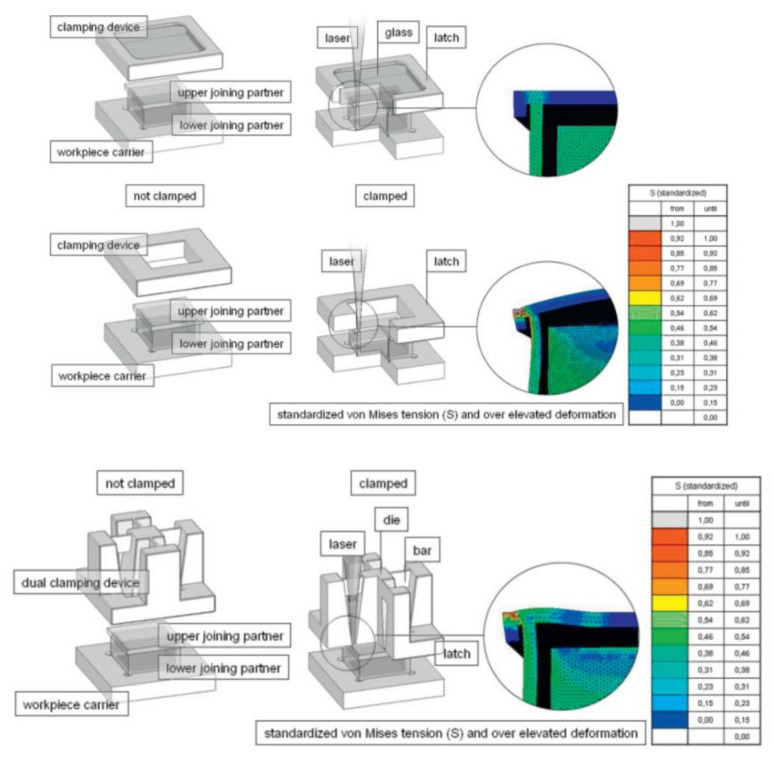
Schematic sketch of LTW showing: (**top**), the clamping unit placed within the beam path (**top**), clamping unit placed not within the beam path (**bottom**) and results of mechanical Finite Element Simulations showing standardised von Mises tensions and over elevated deformations caused by the different clamping techniques (**bottom**) DCD and the results of a mechanical Finite Element Simulation showcasing standardised von Mises tensions and over elevated deformations caused by the new clamping technique [189]. Reproduced with permission from Ref. [189]. Copyright (2013), Physics Procedia.

**Figure 15 polymers-13-00675-f015:**
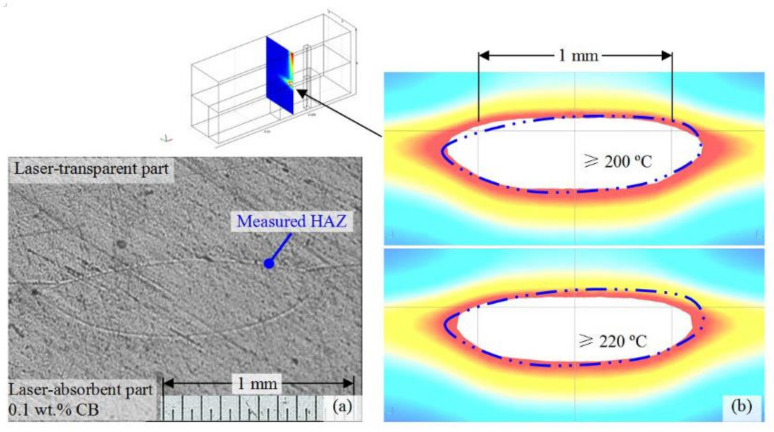
HAZ as observed by a polarized light microscope (**a**) and isotherms obtained from contact model (**b**) for PA6 (0.1 wt.% CB, power 44 W, speed 25 mm/s [203]. Reproduced with permission from Ref. [203]. Copyright (2009), ProQuest Dissertations and Thesis.

**Figure 16 polymers-13-00675-f016:**
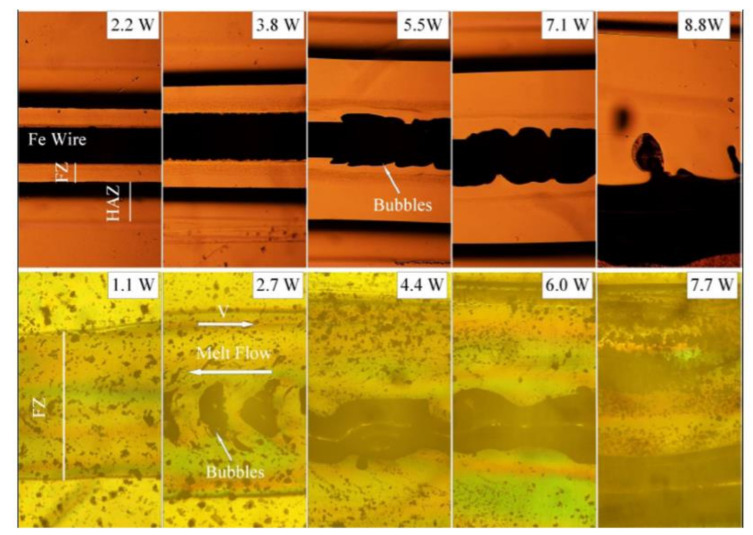
Optical micrographs of joints in Fe wire (**upper**) and CB groups (**lower**) with different input powers(feed speed, V = 1 mm/s and clamping pressure, C = 0.4 MPa) [208]. Reproduced with permission from Ref. [208]. Copyright (2018), Optics and Laser Technology.

**Figure 17 polymers-13-00675-f017:**
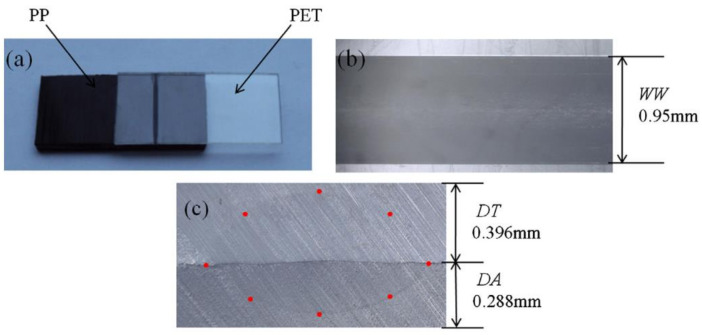
Various images of the weld (**a**) macroscopic welding sample, (**b**) micrograph of the weld width and (**c**) micrograph of the molten pool [183]. Reproduced with permission from Ref. [183]. Copyright (2014), Materials and Design.

**Figure 18 polymers-13-00675-f018:**
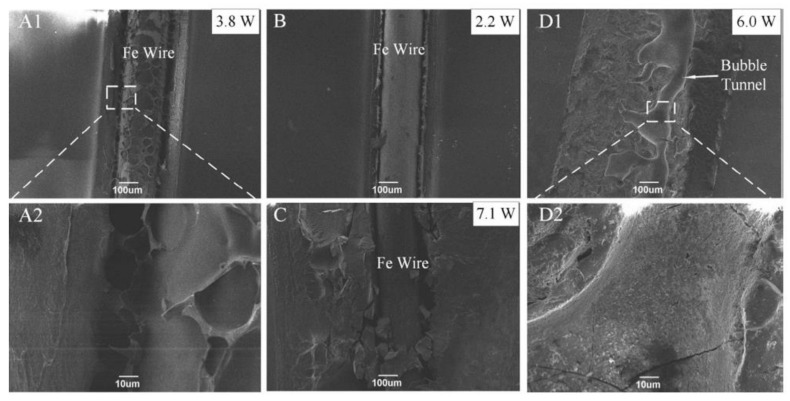
SEM images of welding seams at various sections of Fe wire (**A**–**C**) and CB(**D**) group after undergoing tensile testing with V = 1 mm/s and C = 0.4 MPa [208] Reproduced with permission from Ref. [208]. Copyright (2018), Optics and Laser Technology.

**Figure 19 polymers-13-00675-f019:**
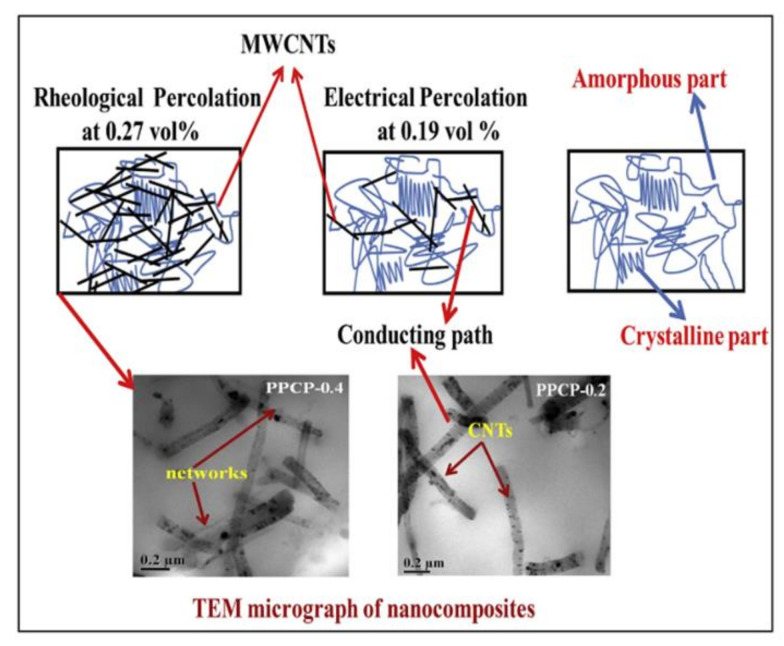
Schematic representation of the rheological and electrical percolation of nanocomposites along with the TEM micrographs [62]. Reproduced with permission from Ref. [62]. Copyright (2016), Polymer Testing.

**Figure 20 polymers-13-00675-f020:**
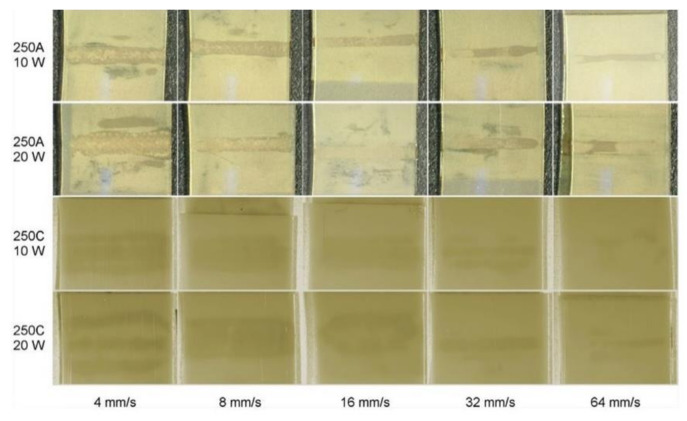
Optical micrographs of the joints before mechanical testing with varying laser power and scanning speed. HAZ disappearing with the increase in scanning speed (64 mm/s). Reproduced with permission from Ref. [10]. Copyright (2010), Materials and Design.

**Figure 21 polymers-13-00675-f021:**
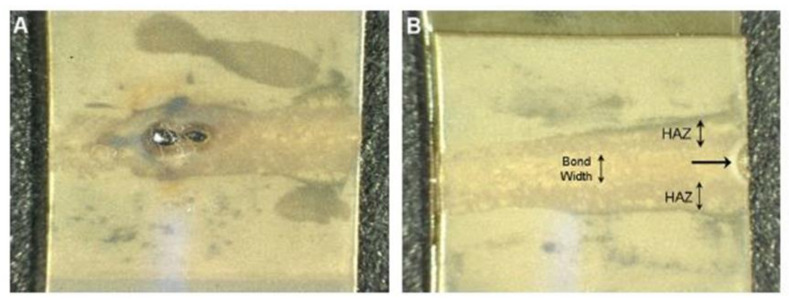
(**A**) Heat damage for the amorphous sample with power 20 W and speed 4 mm/s (**B**) Damage at the joint edge of the sample. Reproduced with permission from Ref. [10]. Copyright (2010), Materials and Design.

**Figure 22 polymers-13-00675-f022:**
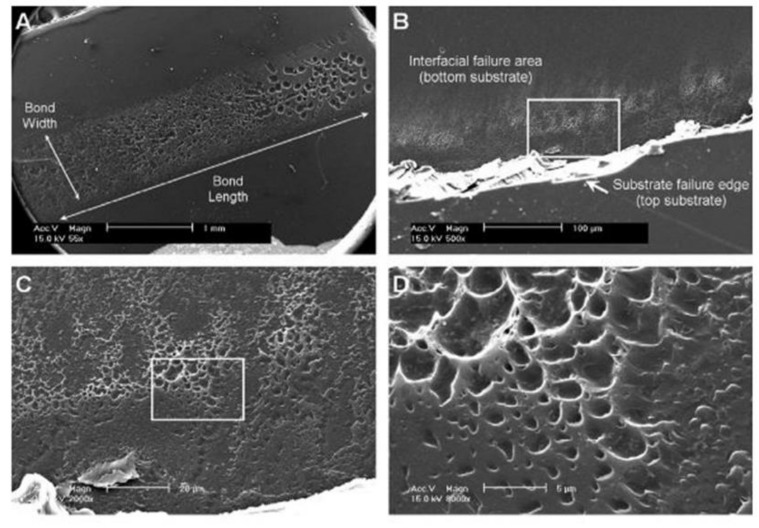
Post-failure SEM images of the weld interface. (**A**) Interfacial failure of the joint in the amorphous sample (**B**) Mixed failure mode in SC sample. (**C**) View of inset shown in (B). (**D**) View of inset shown in (C). Reproduced with permission from Ref. [10]. Copyright (2010), Materials and Design.

**Table 1 polymers-13-00675-t001:** List of LTW pre-and post-processing techniques for the analysis of SC polymers.

Type of Characterisation	Techniques	Analysis
Rheology	Melt flow index (MFI)	Ease of flow of the polymer melt
Thermal Characterisation	Differential scanning calorimetry (DSC) [60]	Phase transformation, Glass transition (T_g_), crystallisation temperature (T_c_) and melt temperature (T_m_)
	Thermo-gravimetric analysis (TGA)	Material degradation temperature
Morphological Analysis	X-ray diffraction (XRD) [61]	Phase information, crystallographic information, residual strain
	Transmission electron microscopy (TEM) [62]	Grain size, microstructure, orientation, diffraction pattern
	Scanning electron microscopy (SEM) [45] and field emission (FE) [51]	Microstructure, topography, grain size, fracture analysis, Weld defects: voids, cracks, porosity, etc.
	Optical microscope [20]	
	Electron probe X-ray micro analyser (EPMA) [63]	Elemental analysis in the weld zone
	Attenuated total reflectance Fourier Transform Infrared Spectroscopy (FT-IR: ATR) and Raman Spectroscopy [64]	Vibrational analysis, Chemical structure, polymer degradation, structural fingerprints
	Wavelength-energy dispersive X-ray spectrometric analysis (WDS, EDS) [65]	Elemental analysis
	Electron back scattering diffraction (EBSD)	Orientation and size of the grain
Mechanical Testing	Vickers Hardness [66,67,68,69]	Micro-hardness measurement
	Universal Testing Machine (UTM) [25]	Lap-shear test,
	Microtome [34]	Slicing thin samples

**Table 2 polymers-13-00675-t002:** Temperature profiles of CFRP with fibre orientation.

Laser Power (W)	Fibre Arrangement	Maximum Temperature (°C)
4	Centred PPS depot	280
	Complete transverse fibre arrangement	360
5	Centred PPS depot	360
	Complete transverse fibre arrangement	>440

**Table 3 polymers-13-00675-t003:** MFI values for polypropylene samples with molecular weight [190].

PP Samples	MFI
High Molecular weight	
Narrow	4.2
Regular-broad	5.0
Broad-regular	3.7
Middle molecular weight	
Narrow	11.6
Regular	12.4
Broad	11.0
Low Molecular weight	
Narrow	25.0
Regular-narrow	23.0

**Table 4 polymers-13-00675-t004:** Weldability matrix from LPKF Laser and Electronics AG 1 = good welded joint, 2/3 = satisfactory welded joint, 1/3 = poor welded joint and 0 = no welded joint [192]. High-density polyethylene (HDPE), PP, poly(methyl methacrylate) (PMMA), polystyrene (PS), polybutylene terephthalate (PBT) and polycarbonate (PC).

		Laser Transmissive Layer					
		HDPE	PP	PMMA	PS	PBT	PC
Laser Absorptive layer	HDPE	1	0	1/3	0	0	0
	PP	0	1	1/3	0	0	0
	PMMA	1/3	1/3	1	2/3	1/3	0
	PS	0	0	2/3	1	2/3	1/3
	PBT	0	0	0	0	1	1
	PC	0	0	1	1/3	1	1

**Table 5 polymers-13-00675-t005:** Various modes of failure in the case of lap joint geometry [10]. Reproduced with permission from Ref. [10]. Copyright (2010), Materials and Design.

Mechanism	Description	Appearance	Inference	Details
Interfacial	Interfacial failure	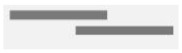	Bond strength << substrate strength	Interfacial failure at bond. Most undesired result
Substrate (Type I)	Bulk substrate failure	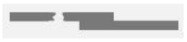	Bond strength >> substrate strength	Substrate results in tensile yield and break. This implies that the joint is as strong as possible, where the substrate will fail first. Most desired result.
Substrate (Type II)	Near interfacial substrate failure	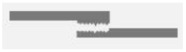	Bond strength > substrate strength	Failure within the substrate, but near the interfacial region and bond still intact.
Mixed (Type I)	Substrate and interfacial	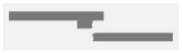	Bond strength ~ substrate strength	Failure partially within substrate and at the bond interface.
Mixed (Type II)	Interfacial and some substrate	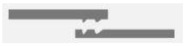	Bond strength < substrate strength	Failure mostly interfacial. but with some substrate failure, e.g., in the form of plastic deformation

## Data Availability

Data sharing not applicable.

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
