# Peer review of "Laser Transmission Welding of Semi-Crystalline Polymers and Their Composites: A Critical Review"

_polymers, 2021, doi:10.3390/polym13050675_

Round 1

Reviewer 1 Report

The manuscript is interesting and informative for the early career researchers as well as seasoned ones. The authors presented a comprehensive review on the joining of semicrystalline polymers using a laser welding approach.
The manuscript is well written and the topic is of current interest for many research groups and industries.

Before accepting this manuscript, the following changes are required.

Firstly, there are a few missing punctuation marks and grammar issues. A quick grammar check will resolve these issues.

There are several figure references missing in the manuscript. For eg: Line 66, 78, 99, 120, 199,

This is usually a word processor issue. Authors should be able to resolve it easily.

Line 118: is there any reference for this statement?
https://www.sciencedirect.com/science/article/abs/pii/S1359836814003606
maybe this one is suitable?

1. There are several missing references to figures in the text. It could you the fault of your word processor
A quick check will solve those issues.

2. Lines 134-136 have already been introduced in the paragraphs above. The repetition is not necessary.

4. Line 201: --> regions. Srinivasan

5. Line 211: Laser radiations? LASER itself contains radiation in its name. May be Laser rays?
Line 217: Domain names should be in italic or bold, to facilitate easier understanding.

6. Provide a list of symbols and abbrevations.

7. Line 296: Just Brunauer Emmett and Teller will be sufficient

Lines 301-303 : reference for tensile properties?

Figure 6: I do not see the interest in showing the aggregate's schematic. It is usually well known.

Section 3.1.3: There are no insights into Laser welding processes. Just the properties of the nanoparticles are mentioned.
Similar to the hybrid filler section, authors should add a few sentences which would infer how these particles affect the welding process.

Line 598: Which force? Correct jargon should be used to avoid confusion. I am assuming clamping force?

Section 4 is unclear. The authors talked about polymer processing a lot but a little has been emphasized on how those processing parameters affect LW.

Table 3: Check the names and labels. They are not correct.

Through-out the manuscript: T_g and T_m.

It is interesting work but why haven't the authors discussed the LW of fiber-reinforced composites?

Reviewer 2 Report

This review focuses on laser transmission welding of polymers, in particular semicrystalline ones and among these, polypropilene, and composites with special interest in carbon black. The technique of laser transmission welding is explained and relevant parameters with influence in the performance are presented and reviewed. The information given in this work may be of interest for the scientific community, and most important from the application point of view, since joining of polymers is still challenging. The document is well structured although at some points English sentences construction is a bit awkward and revision is needed. In addition to this (which must be improved/clarified) there are some additional points that need to be corrected/clarified:

  1. There is a problem with some references: “Error! Reference source not found” appears several times in the manuscript so it is not possible to see the corresponding reference.
  2. In figure 2, in the way it is presented it is impossible to read anything
  3. At several points the expression “laser radiations” is used but in some of the cases it would be more correct to use “laser beam”
  4. Left panel of figure 5 is redundant with figure 4. They might be merged or combined. Also in figure 5 the text corresponding to zone 5 cannot be read easily
  5. In line 203 there is a mention to ablation processes. It might be appropriate to clarify that in the present case ablation is not occurring
  6. In line 247 it is mentioned that the temperature of the laser radiation changes but what is changed is the temperature of the material
  7. The paragraph from line 453 to line 466 describes the role of CB as absorbing component. This section is focused on “hybrid fillers” but in this paragraph no additional component is mentioned besides CB
  8. In lines 485 and 486 it is mentioned that ·when the laser passes through the laser transmissive layer of the polymer, as illustrated in Figure 9 some of its energy is lost due to absorption and/or scattering by the compounding ingredients like fillers, pigments and the crystal structure” It is clear that crystalline structure and/or additives could have effect but what would be the point of having pigments in a layer which should be transparent to the radiation used?
  9. Is not figure 10 redundant with figure 5?
  10. Lines 569-572: The sentence “when laser radiations are emitted on the polymer, it undergoes recrystallization” should be rephrased. First, the radiation is emitted from the laser and it reaches the polymer, or it impinges on the polymer. And second, regarding the recrystallization, this is not a general observation upon irradiation of polymers using lasers. It will depend on the nature of the laser (pulsed vs. CW and pulse duration). In fact when short or ultrashort laser pulses are used, heating takes place very fast and so the cooling does and the polymer does not recrystallize in such a short time scale. It was mentioned in line 96 that the main focus of this work are diode lasers. Then, emphasize here that CW lasers are used. Also, related to this point it would be important to mention what kind of laser is used in the different works reported and cited. In fact, in general along the document information on powers and on times is given but there is no information on wavelengths (and considering optical absorption of polymer materials this is an important (relevant) parameter).
  11. Also, at some points the manuscript seems a bit repetitive. For instance, in line 1186 authors define what HAZ is, but they already wrote about HAZ several pages above.

Reviewer 3 Report

Reviewers' comments:

Manuscript number: polymers-1091719

Title: Laser Transmission Welding of Semi-Crystalline Polymers and their Composites: A Critical Review.

The manuscript needs a detailed editing. It cannot be recommended for publication in the present form. I hope the following points would be helpful for the authors.

The authors need to consider the following comments

1) Keywords: add more keywords.

2) Check line number 66 - …….work of Industry 4.0 technologies, as shown in Error! Reference source not found….

3) Figures 1 and 2 - not clear.

4) Several faults: are added or missing spaces between words.

5) Check line number 99 - Several factors are influencing the quality of weld as shown in Error! Reference source not found…….

6) Figure 3. Not clear make clear.

7) Please provides the references for equations and formula.

8) 3.1.3 Boron Nitride (BN) - should be provide more details.

9) Some sentences need reconstruction and the level of English should be improved.

10) Figure 9. Not clear make clear.

11) Figure 12. Not clear make clear.

12) Conclusion and Future Trends should be specific short results.

13) References: make all references in same format for volume number, page number and journal name, because it is difficult to searching and reading.

14) Author should provide Graphical Abstract.

So that I recommended this manuscript to major revision and for future process.

Round 2

Reviewer 2 Report

The authors have modified the manuscript according to the comments. Please read and check carefeully English style for the final version

Reviewer 3 Report

Reviewers' comments:

The authors revised the manuscript according to the reviewers' comments.